# EFHD2 suppresses intestinal inflammation by blocking intestinal epithelial cell TNFR1 internalization and cell death

Jiacheng Wu [1,5], Xiaoqing Xu[1,5], Jiaqi Duan[1], Yangyang Chai[1], Jiaying Song[1], Dongsheng Gong[1], Bingjing Wang[1], Ye Hu[1,2], Taotao Han[3], Yuanyuan Ding[1,4], Yin Liu[1], Jingnan Li[3] ✉ & Xuetao Cao [1,2] ✉

TNF acts as one pathogenic driver for inducing intestinal epithelial cell (IEC) death and substantial intestinal inflammation. How the IEC death is regulated to physiologically prevent intestinal inflammation needs further investigation. Here, we report that EF-hand domain-containing protein D2 (EFHD2), highly expressed in normal intestine tissues but decreased in intestinal biopsy samples of ulcerative colitis patients, protects intestinal epithelium from TNF-induced IEC apoptosis. EFHD2 inhibits TNF-induced apoptosis in primary IECs and intestinal organoids (enteroids). Mice deficient of *Efhd2* in IECs exhibit excessive IEC death and exacerbated experimental colitis. Mechanistically, EFHD2 interacts with Cofilin and suppresses Cofilin phosphorylation, thus blocking TNF receptor I (TNFR1) internalization to inhibit IEC apoptosis and consequently protecting intestine from inflammation. Our findings deepen the understanding of EFHD2 as the key regulator of membrane receptor trafficking, providing insight into death receptor signals and autoinflammatory diseases.

Inflammatory bowel disease (IBD), including Crohn's disease (CD) and ulcerative colitis (UC), is a group of chronic inflammatory diseases of the gastrointestinal tract characterized by the disrupted intestinal barrier and the severe intestinal inflammatory response[1–3]. The intact epithelium and mucosal barrier jointly maintain the integrity of intestinal barrier and colonic homeostasis. However, excessive intestinal epithelial cell (IEC) death in IBD conversely aggravates the defects and disorders of the epithelium barrier, leading to substantial and persistent intestinal inflammation, which eventually advances the intestinal carcinogenesis[3,4].

As a pleiotropic pro-inflammatory cytokine, TNF has been considered a primary driver of epithelium cell death in the intestine, and TNF blockade has achieved clinical benefit for IBD patients in clinic[5,6].

Upon TNF stimulation, TNFR1 normally recruits TRADD and RIPK1 to form complex I on the plasma membrane, which mediates NF-κB signaling. Once TNFR1-complex I is internalized by a clathrin-mediated endocytosis, de-ubiquitylation and degradation of RIPK1 lead to the dissociation of complex I and the formation of complex II, which induces apoptosis eventually[7–9]. A recent study has characterized ATG9A as a checkpoint of TNF-induced apoptosis by targeting LC3-independent lysosomal degradation of cytotoxic complex IIa[10]. Therefore, receptor internalization and endosomal trafficking are essential in the initiation and termination of TNFR1-mediated cell death signaling[8]. Indeed, membrane translocation and endosomal trafficking of immune receptors for cytokines, chemokines or cell death factors are verified to be important in various biological

[1]Department of Immunology, Institute of Basic Medical Sciences, Peking Union Medical College, Chinese Academy of Medical Sciences, Beijing 100005, China. [2]Frontier Research Center for Cell Response, Institute of Immunology, College of Life Sciences, Nankai University, Tianjin 300071, China. [3]Department of Gastroenterology, Key Laboratory of Gut Microbiota Translational Medicine Research, Peking Union Medical College Hospital, Chinese Academy of Medical Sciences, Beijing 100730, China. [4]Suzhou Institute of Systems Medicine, Chinese Academy of Medical Sciences, Suzhou 215123, China. [5]These authors contributed equally: Jiacheng Wu, Xiaoqing Xu. ✉e-mail: lijn@pumch.cn; caoxt@immunol.org

processes, but still underrated in the control of immune response and inflammation. Another study revealed the mechanism of spatio-temporal regulation of IFNAR endocytosis and endosomal sorting to differential activation of JAK-STAT signaling by IFN-α and IFN-β[11]. However, whether there exists suppressive molecular machinery in the membrane trafficking of the cell death receptors such as TNFR1 in IECs remains to be fully investigated.

EF-hand domain-containing protein D2 (also known as Swiprosin-1) is a calcium-binding protein with two conserved EF-chiral coiled-coil structural domains, and it is mostly involved in calcium signaling, cytoskeleton assembly and synapse formation[12–14]. EFHD2 is broadly expressed in various cell types, with particularly high expression in neurons[12]. The studies of the pathological role of EFHD2 therefore have focused on behavioral pathologies, neurological disorders, and neuro-degenerative diseases[12,15–17], highlighting the role of EFHD2 in regulating the membrane-cytoplasm trafficking of membrane-associated proteins. Our previous study showed that EFHD2 can interact with phosphory-lated IFN-γR2 to promote IFN-γR2 trafficking from Golgi to cell membrane of macrophages upon intracellular bacterial infection, which facilitates subsequent IFN-γ signaling in innate response[18]. Therefore, further exploration for the regulation of membrane receptor trafficking by EFHD2 will better understand the control of immunity and inflammation.

Besides immune and neural tissues, *Efhd2* is also abundantly expressed in colon and duodenum[13,14]. Our preliminary experiments indicate the relatively high expression of EFHD2 in IECs. Therefore, we wonder the potential function of EFHD2 in the intestinal pathological processes. In this study, we identify EFHD2 as an endogenous sup-pressor of epithelial cell death by restraining the endocytosis of TNFR1 to suppress TNF-mediated death signaling. We find that EFHD2 can inhibit TNF-induced apoptosis in primary IEC, intestinal organoids (enteroids) and intestinal epithelial cell lines. Deficiency of EFHD2 in mouse IECs causes more severe intestinal symptoms in experimental colitis in vivo.

## Results

### *Efhd2* deficiency exacerbates experimental colitis

Following our investigation of Efhd2 in the macrophage innate response[18], we examined the profile of Efhd2 protein expression in mouse organs, and we found the relatively higher level of Efhd2 expression in normal intestine and brain (Supplementary Fig. 1a). However, we detected the lower expression of EFHD2 protein in UC patients or mouse colon tissues with dextran sodium sulfate (DSS)-induced experimental colitis (Fig. 1a, b). The immunostaining showed that the EFHD2 expression in intestinal biopsy samples was sig-nificantly decreased in UC patients at the active stage compared with that in UC patients at the remission stage and in healthy controls (Fig. 1a, c). Similarly, the expression of Efhd2 in the colon of mice with DSS-induced colitis was reduced at the period of active acute inflam-mation on day 7, and then restored to be comparable to that of untreated controls at the period of remission on day 12 (Fig. 1b, c). EFHD2 was mainly expressed in the cytoplasm of epithelium in colon tissues, with relatively lower expression in the lamina propria (Fig. 1a, b). Likewise, by analyzing public single-cell RNA sequencing data[19], we found that *EFHD2* was highly expressed in various epithelial cells in normal human colon tissue (Supplementary Fig. 1b). Therefore, our results implicated that decreased EFHD2 expression in intestinal epithelium might be involved in intestinal inflammation.

To further investigate the role of *Efhd2* in intestinal inflammation, we firstly constructed *Efhd2*[-/-] mice by crossing *Efhd2*[f/f] mice carrying loxp sites flanking the second exon of *Efhd2* with 129S/Sv-Tg (Prm-cre) 58Og/J mice[18] (Supplementary Fig. 2a–c). *Efhd2* deficiency did not affect the intestinal development under the steady-state condition (Supplementary Fig. 2d, e). However, after challenging with 3% DSS, *Efhd2*[-/-] mice developed more exacerbated colitis, manifested with

increased loss of body weight, higher disease activity index (DAI) score and reduced length of colons, compared with similarly treated WT littermate controls (Fig. 1d–f). Correspondingly, *Efhd2*[-/-] mice showed more severe disruption of the intestinal mucosal epithelium barrier accompanied by increased inflammatory cell infiltration and decreased intestinal goblet cells (Fig. 1g, h). Besides, an increased proportion of granulocytes and CD4[+] T cells were recruited to lamina propria of *Efhd2*[-/-] mice after DSS administration (Supplementary Fig. 2f, g).

Together, our findings suggested that EFHD2 expression was decreased in the autoimmune inflammatory intestine and *Efhd2*-defi-cient mice were more sensitive to experimental acute colitis.

### Intestinal epithelial deficiency of *Efhd2* exacerbates experi-mental colitis

To examine the possible role of *Efhd2* in myeloid cells in the regulation of colitis, we generated mice with conditional knockout of *Efhd2* in myeloid cells (*Efhd2*[f/f] Lyz[cre/+]) by crossing *Efhd2*[f/f] mice with Lyz-Cre mice. Following the same procedure of DSS administration, we found no difference in the body weight loss, DAI score, colon length (Sup-plementary Fig. 3a–c) or histological morphology of colon tissues (Supplementary Fig. 3d) between *Efhd2*[f/f] Lyz[cre/+] and *Efhd2*[f/f] mice. These results excluded the importance of *Efhd2* in myeloid cells during DSS-induced intestinal inflammation.

We then generated mice with conditional knock-out of *Efhd2* in the intestinal epithelium (*Efhd2*[f/f] Vil1[cre/+]) by crossing *Efhd2*[f/f] mice with Vil1-Cre mice (Supplementary Fig. 3e–g). Under the steady-state con-dition, deletion of *Efhd2* in IECs of mice did not affect the length of colons (Supplementary Fig. 3h). Additionally, we conducted immu-nostaining of intestinal tissues including Ki67 to detect the prolifera-tion of intestinal epithelium, and multiple markers of different IEC subtypes of small intestine and colon, such as Lysozyme for Paneth cells, Chromagranin-A for enteroendocrine cells, Carbonic Anhydrase 1 (CA1) for colonocytes, Muc-2 and PAS for goblet cells[20]. Indeed, we found that *Efhd2* deficiency in IECs did not affect intestinal develop-ment, differentiation and proliferation (Supplementary Fig. 3i, j). However, *Efhd2*[f/f] Vil1[cre/+] mice were more susceptible to DSS-induced colitis than *Efhd2*[f/f] mice, characterized by significantly more body weight loss, higher DAI score and shorter colon lengths (Fig. 2a–c). Histology of colon tissues from *Efhd2*[f/f] Vil1[cre/+] mice showed more severe damage of intestinal epithelium accompanied with the increased inflammatory cell infiltration and goblet cells loss (Fig. 2d–f) and increased production of proinflammatory cytokine IL-6 and IL-1 (Fig. 2g). Moreover, we observed significantly reduced staining of the epithelial marker E-cadherin in distal colon tissues of *Efhd2*[f/f] Vil1[cre/+] mice relative to controls after DSS treatment (Fig. 2h). Correspond-ingly, the intestinal permeability assay showed higher permeability of the intestinal barrier in *Efhd2*[f/f] Vil1[cre/+] mice than that in *Efhd2*[f/f] mice (Fig. 2i). Above results suggested an obvious decrease of IECs and impaired intestinal barrier upon DSS treatment. Thus, deficiency of *Efhd2* in intestinal epithelium exacerbated mouse acute colitis and intestinal epithelial barrier damage.

### Efhd2 inhibits intestinal epithelial cell apoptosis during experi-mental colitis induction

To investigate how *Efhd2* deficiency enhanced the damage of intestinal epithelium, we examined cell death in colonic tissue sections by terminal deoxynucleotidyl transferase dUTP nick end labeling (TUNEL) staining and found increased number of TUNEL[+] IECs in *Efhd2*[f/f] Vil1[cre/+] mice relative to controls upon DSS treatment (Fig. 3a). The results suggested that Efhd2 in IECs may maintain the integrity of the intest-inal epithelial barrier by inhibiting intestinal epithelial cell death.

Apoptosis and necroptosis are main programmed cell death pathways of IECs in colitis[4]. Both excessive apoptosis and necrosis lead to the breakdown of the intestinal barrier, which in turn exacerbates

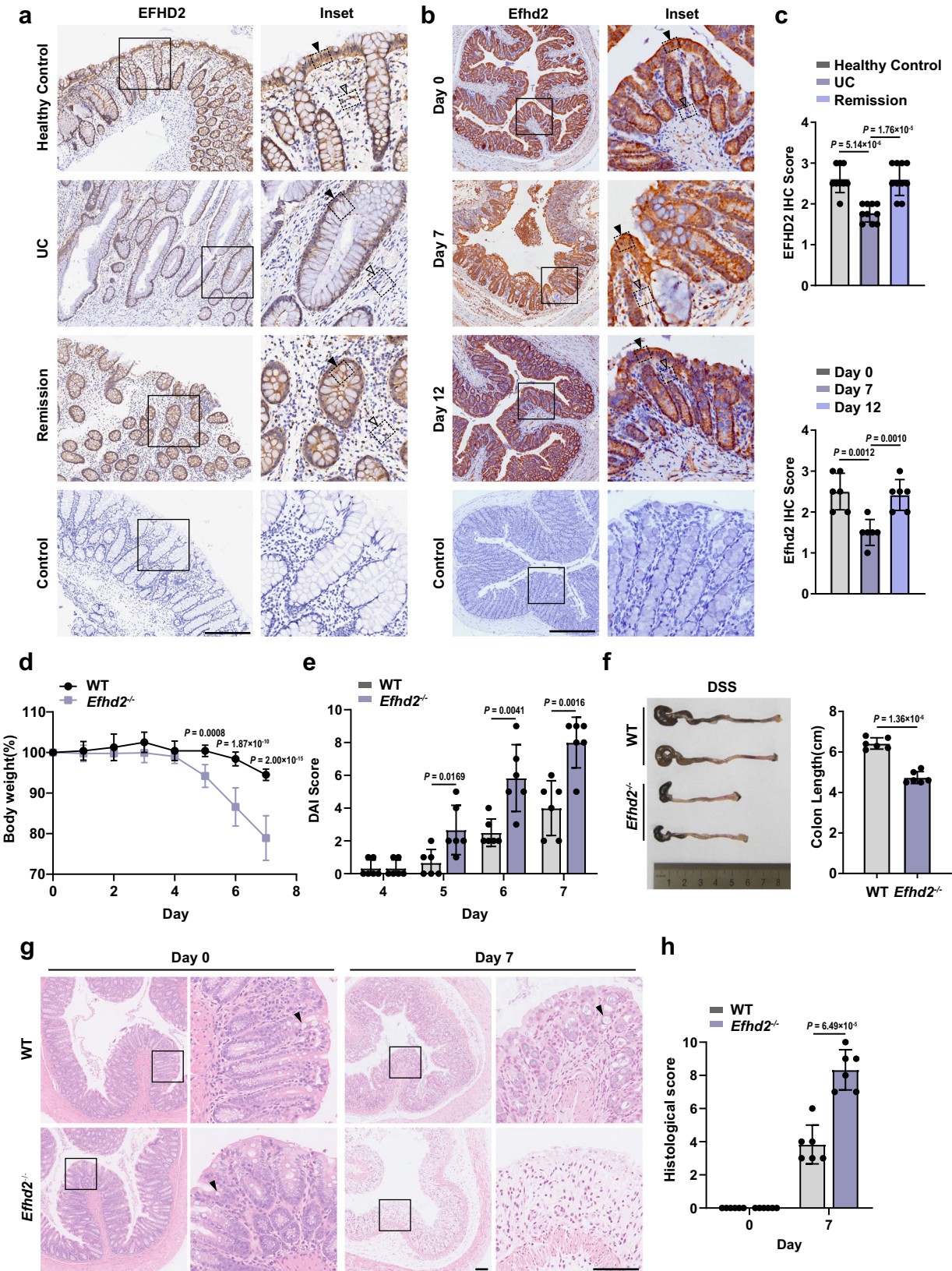

bacterial invasion and inflammatory response in the intestine[4,21,22]. We performed immunofluorescence staining for cleaved caspase-3 and cleaved caspase-8 (markers of apoptosis) and for phosphorylated mixed-lineage kinase domain-like protein (p-MLKL, a marker of necroptosis[23]) in the colon of *Efhd2*^f/f and *Efhd2*^f/f Vil1^cre/+ mice after DSS stimulation. We found that compared to those in the intestinal

epithelial villi of *Efhd2*^f/f mice, levels of cleaved caspase-3 and cleaved caspase-8, which were mainly distributed in the proximal and mid colonic regions, were significantly higher in *Efhd2*^f/f Vil1^cre/+ mice, whereas the p-MLKL level was similar between *Efhd2*^f/f Vil1^cre/+ and *Efhd2*^f/f mice (Fig. 3b, c). Besides, the elevated IEC apoptosis observed in the *Efhd2*^f/f Vil1^cre/+ mice was primarily localized to the top regions of

**Fig. 1 | *Efhd2* deficiency exacerbates DSS-induced colitis. a** Representative images of immunostaining for EFHD2 on intestinal biopsy samples from ulcerative colitis (UC) patients at the active or the remission stage, healthy controls and controls stained only with the secondary antibody. Scale bar, 300 μm. Solid arrowheads show intestinal epithelium and hollow arrowheads show the lamina propria. **b** Representative images of immunostaining for Efhd2 on distal colon tissues from C57BL/6J mice treated with 3% dextran sodium sulfate (DSS) for 6 days and sacrificed at the indicated time with secondary antibody controls. Scale bar, 400 μm. Solid arrowheads show intestinal epithelium and hollow arrowheads show the lamina propria. **c** Histologic scores of EFHD2 (**a**) and Efhd2 (**b**) staining were using a 3-point quantification scale. UC patients at the active or the remission stage (*n* = 10 each group), healthy controls (*n* = 9) in **a**. *n* = 6 mice each group in **b**. Relative percentage of body weight change (**d**), disease activity index (DAI) (**e**), and representative images and lengths of the colon (**f**) of WT and *Efhd2*-/- mice after 3% DSS administration for 6 days and sacrificed on day 7. Representative images of hematoxylin and eosin (H&E) staining of distal colon (**g**) and histologic scores (**h**) of WT and *Efhd2*-/- mice subjected to DSS-induced colitis on day 0 and day 7. Left scale bar, 100 μm; right scale bar, 200 μm; arrows show intestinal goblet cells in **g**. *n* = 6 mice each group (**d**–**f**, **h**). Data are representative of three independent experiments; error bars show means ± s.d. *P* values were determined by unpaired two-tailed *t*-test. For body weight curves, two-way ANOVA analysis with Sidak's multiple comparisons test. Source data are provided as a Source Data file.

villi in the intestinal epithelium, while a small portion occurred within the midsection of the crypt axis (Fig. 3b). We then isolated primary IECs from these mice after DSS stimulation and found, consistently, a significantly increased cleaved caspase-3 and cleaved caspase-8 level, but not the p-MLKL level, in IECs of *Efhd2*f/f Vil1cre/+ mice (Fig. 3d). These results indicated that Efhd2 has a critical role in inducing IEC apoptosis but not necroptosis during experimental colitis induction.

We also performed the transcriptome analysis of primary IECs from mice after DSS administration on day 7. It revealed that genes with increased expression in *Efhd2*-/- IECs compared to controls were associated with the acute inflammatory response, neutrophil and myeloid leukocyte migration (Fig. 3e and Supplementary Fig. 4a, b), consistent with the earlier results of increased immune cells infiltration. After DSS stimulation, the expression of genes related to the apoptotic signaling pathway was significantly increased in *Efhd2*-deficient IECs (Fig. 3f), which provide further evidence for the function of Efhd2 on IEC apoptosis.

Taken together, these results suggested that deficiency of *Efhd2* in IECs promoted IEC apoptosis, but not necroptosis, to impair the integrity of intestinal barrier and therefore caused the more severe colitis in mice induced by DSS.

## Efhd2 protects intestinal epithelium from TNF-induced apoptosis

To investigate whether Efhd2 in IECs exerted an inhibitory effect on TNF-induced apoptosis, we intravenously injected recombinant mouse TNF to *Efhd2*f/f and *Efhd2*f/f Vil1cre/+ mice to induce rapid IEC death, mainly at the villi tips of the small intestine[24]. We found the increased IEC death in the intestinal barrier of *Efhd2*f/f Vil1cre/+ mice, as evidenced by a significantly increased level of cleaved caspase-3 staining in small intestine villi relative to controls (Fig. 4a). Accordingly, shortened villi and more severe damage of intestinal barrier were observed in IEC *Efhd2*-deficient mice (Fig. 4b). These data indicated that Efhd2 protected the intestinal epithelium from TNF-induced apoptosis in vivo.

Furthermore, we isolated intestinal crypts from *Efhd2*f/f and *Efhd2*f/f Vil1cre/+ mice and cultured primary enteroids (Supplementary Fig. 5a). The viability and buds were similar between WT and *Efhd2*-/- enteroids (Supplementary Fig. 5b, c). The immunostaining for Edu or multiple markers for Paneth cell (Lysozyme) and goblet cell (Muc-2) showed the similar development, differentiation and proliferation between WT and *Efhd2*-/- enteroids (Supplementary Fig. 5d). Besides, we performed propidium iodide (PI) staining for necrotic cores on WT and *Efhd2*-/- enteroids, and excluded the formation of necrotic cores inside the enteroids (Supplementary Fig. 5e). We subsequently performed TNF-induced apoptosis experiments on mature enteroids. Live imaging revealed increased caspase-3/7 activity in *Efhd2*-/- enteroids compared to control enteroids treated with TNF for 24 h (Fig. 4c). Similar findings were also observed in enteroids treated with TNF and LCL-161 (TS), a second mitochondrial activator of caspase (SMAC) mimetic to induce RIPK1-dependent apoptosis or with TNF and Cycloheximide (CHX) (TC) to induce RIPK1-independent apoptosis (Supplementary Fig. 5f). Consistently, *Efhd2*-/- enteroids showed higher activity of caspase-3/7

after TNF stimulation by flow cytometry analysis (Fig. 4d and Supplementary Fig. 5g). However, there was no significant difference in the PI intensity between *Efhd2*-/- and WT enteroids when we induced necroptosis by combining zVAD-fmk, a pan-caspase inhibitor, with either TNF and CHX (TCZ) or TNF and LCL-161 (TSZ) (Supplementary Fig. 5h). Furthermore, we treated enteroids with TNF in combination with GSK'872 (inhibitor for RIPK-3) or Z-IETD-FMK (specific inhibitor for caspase-8). We found the increased cell death of *Efhd2*-/- enteroids indicated by PI which could only be rescued by GSK'872 combined Z-IETD-FMK but not by GSK'872 only (Fig. 4e), indicating that Efhd2 could protect enteroids from TNF-induced apoptosis but not necroptosis.

To further rule out additional sensitivity to TNF-independent death pathways in vivo, we treated *Efhd2*f/f and *Efhd2*f/f Vil1cre/+ mice with anti-TNF antibody or control IgG intraperitoneally every 2 days since DSS administration. Compared with control group, the symptoms of acute colitis in *Efhd2*f/f Vil1cre/+ mice were obviously alleviated after treated with anti-TNF antibody, manifested as the fewer body weight loss, lower DAI score, longer colon lengths, and milder histologic damage of intestinal epithelium (Fig. 4f–j).

Taken together, our findings demonstrated that *Efhd2* deficiency promotes TNF-induced IEC apoptosis in both mouse model and the enteroids.

## EFHD2 suppresses RIPK1-dependent and -independent TNF-induced apoptosis

To further elucidate the molecular mechanism of EFHD2 in the inhibition of IEC apoptosis, we used human colorectal cancer cell lines HT-29 and HCT-116. Firstly, knocking down *EFHD2* in both cell lines (Supplementary Fig. 6a, b) significantly elevated the cleaved caspase-8, cleaved caspase-7 and cleaved caspase-3 levels after TNF, TS or TC stimulation (Fig. 5a and Supplementary Fig. 6c) but caused no significant difference in levels of p-RIPK1 and p-MLKL induced by TCZ or TSZ (Supplementary Fig. 6d), which implied an increased apoptosis but not necroptosis after silence of *EFHD2*. We then generated *EFHD2*-/- HCT116 and HT29 cell lines by CRISPR/Cas9 (Supplementary Fig. 6e). Consistent with results from knockdown of *EFHD2*, deletion of *EFHD2* resulted in a significantly higher proportion of annexin V+ PI- cells and enhanced caspase-3/7 activity than in controls when treated with TC or TS (Fig. 5b–d and Supplementary Fig. 6f), but comparable proportion of PI+ cells when treated with TSZ (Supplementary Fig. 6h). Furthermore, western blots analysis also showed increased apoptosis (Fig. 5e and Supplementary Fig. 6g) but comparable necroptosis (Fig. 5e) in *EFHD2*-deficient cells under indicated TNF treatments, accordant with our findings in mice enteroids. Collectively, above findings demonstrated that EFHD2 significantly inhibited TNF-induced RIPK1-dependent and -independent IEC apoptosis.

The binding of TNF with TNFR1 on the plasma membrane initiates the cell death pathway and the NF-κB signaling pathway[5]. When the apoptotic signaling pathway is initiated, RIPK1 is de-ubiquitylated and degraded to turn off TNF-driven NF-κB signaling[5]. We investigated whether EFHD2 affected the activation of NF-κB signaling pathway under TNF stimulation, and we found a significantly decreased phosphorylation of IKKα/β and P65 in *EFHD2*-/- HCT-116 cells compared to

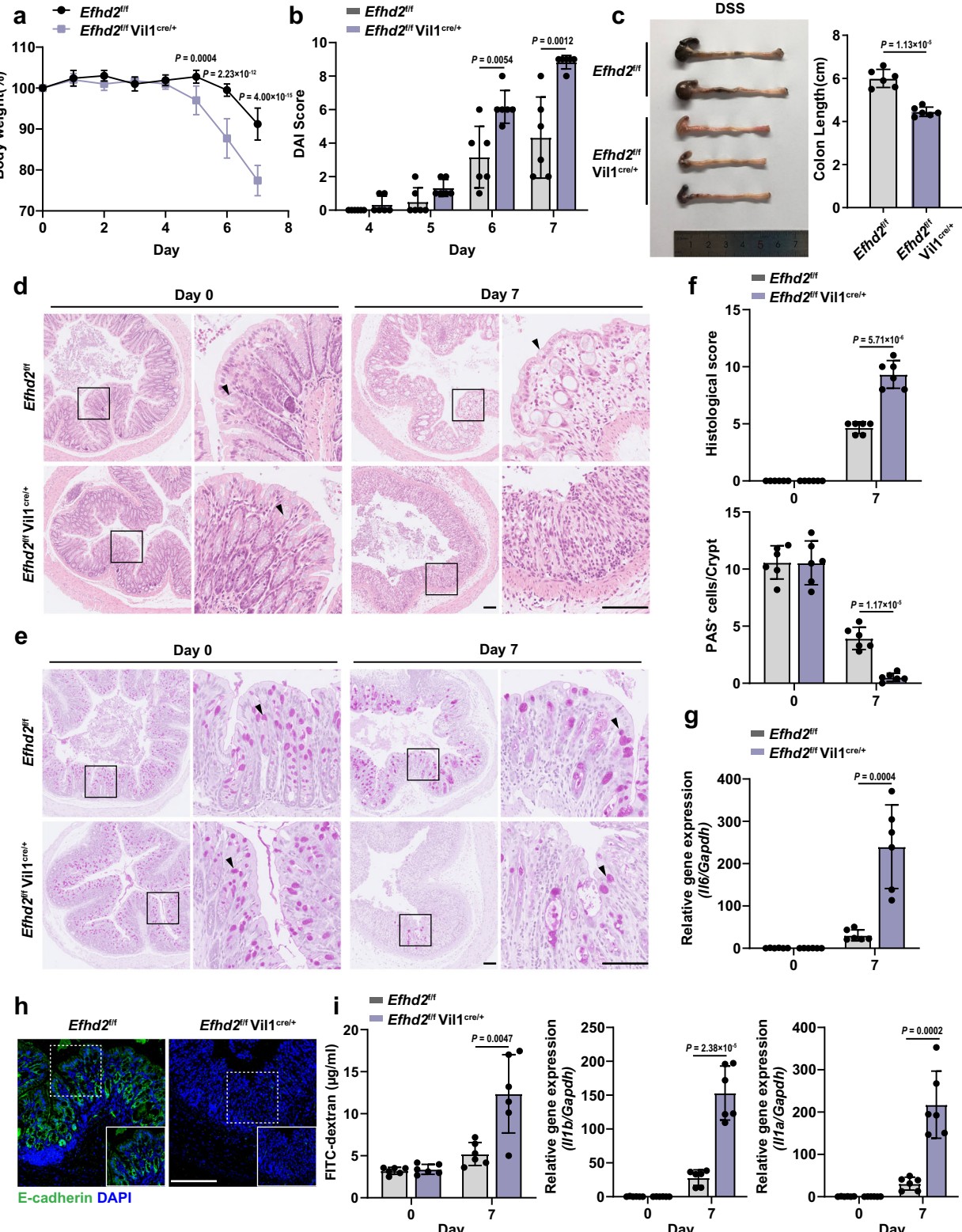

**Fig. 2 | Epithelial *Efhd2* deficiency exacerbates DSS-induced colitis.** Relative percentage change of body weight (**a**), DAI score (**b**), and representative images of colon morphology and lengths (**c**) of *Efhd2*^f/f^ and *Efhd2*^f/f^ Vil1^cre/+^ mice after 2.5% DSS administration for 6 days and sacrificed on day 7. Representative images of H&E (**d**) and periodic acid-Schiff (PAS) (**e**) staining of distal colon of *Efhd2*^f/f^ and *Efhd2*^f/f^ Vil1^cre/+^ mice subjected to DSS-induced colitis on day 0 and day 7. Left scale bar, 100 μm; right scale bar, 200 μm; arrows show intestinal goblet cells. **f** Histologic scores and PAS-positive cells per crypt in **d**, **e**. **g** Relative gene expression levels of *Il6*, *Il1b* and *Il1a* from colonic tissues of *Efhd2*^f/f^ and *Efhd2*^f/f^ Vil1^cre/+^ mice on day 7 after 6-day DSS treatment compared to day 0. **h** Representative images of E-cadherin (green) staining of the distal colon from *Efhd2*^f/f^ and *Efhd2*^f/f^ Vil1^cre/+^ mice after DSS administration on day 7. Scale bar, 150 μm. **i** Concentration of FITC-dextran from *Efhd2*^f/f^ and *Efhd2*^f/f^ Vil1^cre/+^ mice subjected to DSS-induced colitis was measured to detect intestinal permeability on day 0 and day 7. $n = 6$ per group (**a**–**c**, **f**, **g**, **i**). Data are representative of three independent experiments; error bars show means ± s.d. *P* values were determined by unpaired two-tailed *t*-test. For body weight curves, two-way ANOVA analysis with Sidak's multiple comparisons test. Source data are provided as a Source Data file.

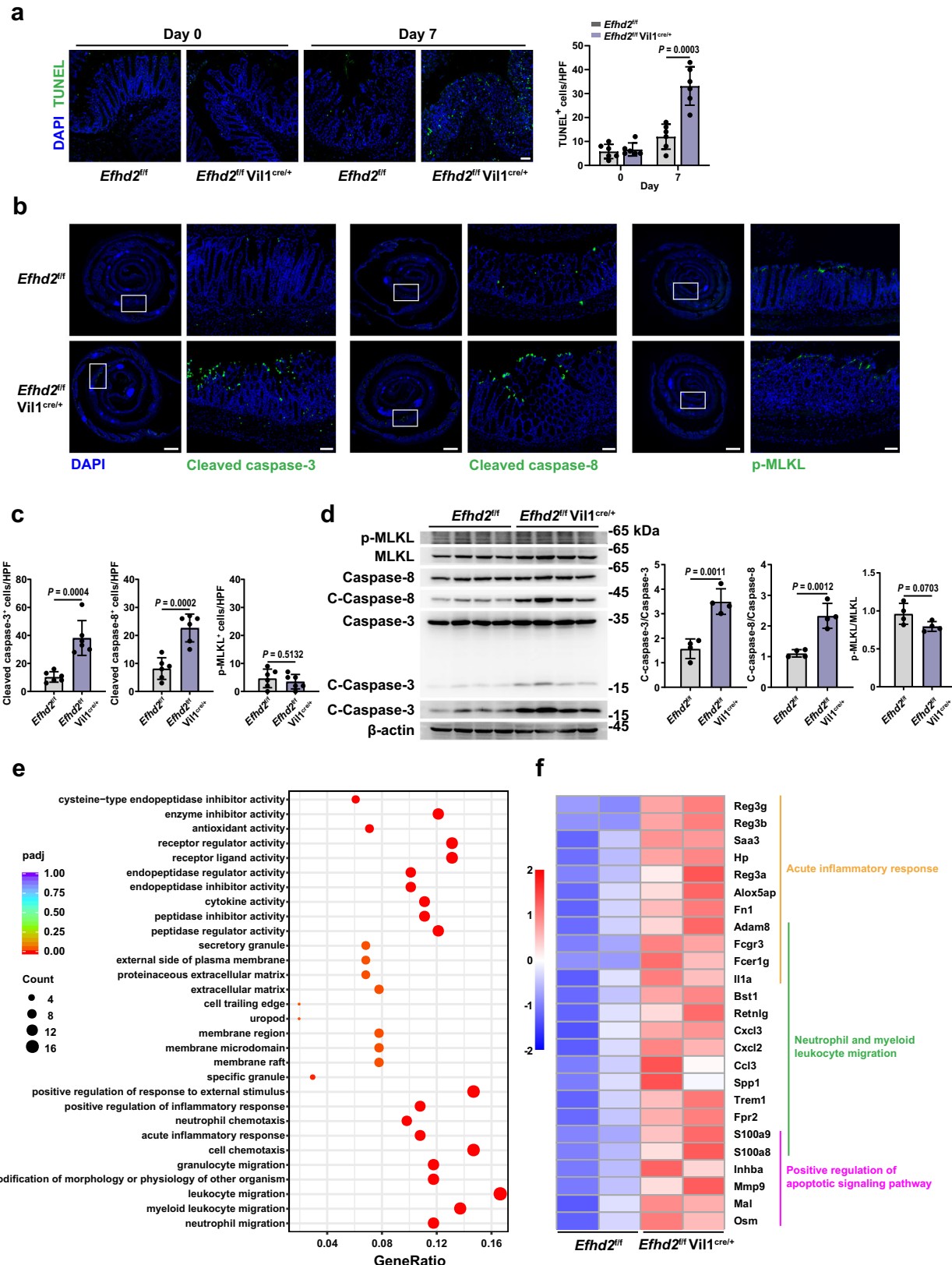

controls stimulated by TNF (Fig. 5f). Likely, the levels of NF-κB targeting genes including cIAPs, cFLIP, Bcl-2 or Bcl-xl were also decreased in *EFHD2*[-/-] HCT-116 cells upon TNF treatment (Supplementary Fig. 6i). The results demonstrated that EFHD2 suppressed TNF-induced apoptosis but promoted NF-κB signaling.

## EFHD2 interacts with complex I to block TNFR1 internalization for inhibiting apoptosis

To gain insight into EFHD2's function on TNF-induced apoptotic signaling pathway, we performed Co-IP analysis and found that the exogenously expressed EFHD2 interacted with RIPK1 as well as with TNFR1

**Fig. 3 | EFHD2 inhibits apoptosis but not necroptosis in intestinal epithelial cells. a** Terminal deoxynucleotidyl transferase dUTP nick end labeling (TUNEL) staining of the distal colon from *Efhd2*[f/f] and *Efhd2*[f/f] Vil1[cre/+] mice and counts of TUNEL-positive cells before and after DSS administration on day 7. Scale bar, 50 μm; n = 6 each group; HPF, high power field. **b** Immunofluorescent detection of cleaved caspase-3, cleaved caspase-8 and p-MLKL (green) in colonic samples from *Efhd2*[f/f] and *Efhd2*[f/f] Vil1[cre/+] mice after DSS administration on day 7. Left scale bars of each marker, 1000 μm; right scale bars of each marker, 100 μm. **c** Quantifications of cleaved caspase-3, cleaved caspase-8 and p-MLKL positive cells from **b**. *n* = 6 each group. **d** Western blot analysis and quantifications of cleaved caspase-3 (C-Caspase-3), cleaved caspase-8 (C-Caspase-8) and p-MLKL in lysates of isolated IECs from DSS-treated *Efhd2*[f/f] and *Efhd2*[f/f] Vil1[cre/+] mice on day 7. *n* = 4 each

group. **e** Gene Ontology (GO) enrichment for biological processes from significantly up-regulated transcripts in the transcriptome of IECs from DSS-treated *Efhd2*[f/f] Vil1[cre/+] versus *Efhd2*[f/f] mice on day 7. GO enrichment analysis of differentially expressed genes was implemented by the clusterProfiler R package (padj <0.05). The padj was calculated with Benjamini-Hochberg multiple testing adjustment. Dot color corresponds to the padj of enrichment, and dot size is determined by the gene count (two-sided). **f** Heatmap of significantly increasingly expressed genes related to acute inflammatory response, neutrophil and myeloid leukocyte migration, and positive regulation of apoptotic signaling pathway from **e**. Two replicates each group. Data are representative of three independent experiments; error bars show means ± s.d. *P* values were determined by unpaired two-tailed *t*-test. Source data are provided as a Source Data file.

and TRADD in complex I upon TNF stimulation (Fig. 6a). The endogenous RIPK1 also interacted with the endogenous EFHD2, which was increased with the TNF treatment (Fig. 6b). The results suggested that EFHD2 might have a role in the initiation period of TNF-apoptotic signaling.

Given that EFHD2 suppressed TNF-induced apoptosis in RIPK1-dependent and -independent way, we hypothesized that EFHD2 interacted with RIPK1 of complex I as a scaffold to modulate the internalization of TNFR1 to regulate apoptosis. Therefore, we examined the cell-surface level of TNFR1 after TNF stimulation and discovered a significantly decreased surface TNFR1 level in *EFHD2*[-/-] HT-29 and HCT-116 cells stimulated with TNF by flow cytometric analysis (Fig. 6c, d and Supplementary Fig. 7a), even at the early stage of TNFR1 endocytosis (Supplementary Fig. 7c), whereas the level of total TNFR1 was not affected (Supplementary Fig. 7b). Thus, EFHD2 might inhibit the apoptotic signaling pathway by suppressing the early endocytosis process of TNFR1. Moreover, immunofluorescence analysis showed more endosomes, represented by EEA1, and an increased number of endosomes co-localized with TNFR1 in *EFHD2*[-/-] HT-29 cells early after TNF stimulation (Fig. 6e). We then isolated early endosomes from WT and *EFHD2*[-/-] HT-29 cells upon TNF stimulation and found significantly higher levels of TNFR1 in early endosomes of *EFHD2*[-/-] HT-29 cells (Fig. 6f). To further verify Efhd2's function in primary IECs, we pretreated WT and *Efhd2*[-/-] enteroids with monodansylcadavrine (MDC), an established inhibitor of TNFR1 internalization, and then treated them with TNF. When the internalization of TNFR1 was restrained, we found that the effect of Efhd2 on TNF-induced apoptosis was counteracted, which was shown by comparable caspase-3/7 activities between WT and *Efhd2*[-/-] enteroids (Fig. 6g and Supplementary Fig. 7d). This result suggested that EFHD2 inhibited apoptosis by blocking TNFR1 internalization. Collectively, EFHD2 interacted with complex I and inhibited the internalization of TNFR1, which ultimately inhibited IEC apoptosis.

## EFHD2 binds Cofilin and inhibits Cofilin phosphorylation to block TNFR1 internalization

To further clarify the mechanism by which EFHD2 inhibited the internalization of TNFR1, we performed Mass Spectrometry (MS) following Co-IP from TNF-treated HCT-116 cells and identified Cofilin as a candidate of EFHD2-associated proteins (Fig. 7a and Supplementary Fig. 8a). Then we confirmed the increased interaction between EFHD2 and Cofilin early after the TNF treatment (Fig. 7b). Cofilin is a key actin depolymerization factor and has been reported to be involved in the dynamic of actin cytoskeleton in host cells during the invasion processes of pathogens, like viruses, bacteria and fungi[25]. We further found that the phosphorylation of Cofilin was enhanced in HCT-116 cells under TNF stimulation after knockdown of *EFHD2* (Fig. 7c).

Since phosphorylation of Cofilin leads to autoinhibition of its activity[26], we examined the relevance of the phosphorylation of Cofilin and TNF-induced apoptosis by pretreating cells with BMS-3, a LIMK inhibitor, to restrain the phosphorylation of Cofilin. We found a dose-dependent decrease of PI[-] caspase-3/7 active HCT-116 cells pretreated

with BMS-3 compared to controls (Fig. 7d), which indicated that phosphorylation of Cofilin promoted TNF-induced apoptosis. Furthermore, when the phosphorylation of Cofilin was restrained by BMS-3, the enhanced caspase-3/7 activity in *EFHD2*[-/-] HT-29 cells was counteracted, as determined by comparable caspase-3/7 activities between WT and *EFHD2*[-/-] HT-29 cells (Fig. 7e), indicating that EFHD2 might suppress the phosphorylation of Cofilin to inhibit cellular apoptosis.

We further determined that TNF plus LCL-161 treatment did not decrease the cell-surface level of TNFR1 in *EFHD2*[-/-] HT-29 cells after inhibiting the phosphorylation of Cofilin by BMS-3 (Fig. 7f), which suggested that the blockade of TNFR1 internalization by EFHD2 depended on keeping the phosphorylation level of Cofilin low by EFHD2.

## Higher EFHD2 expression in intestinal epithelium correlates with better response to anti-TNF therapy in UC patients

Our findings indicated EFHD2 is an endogenous inhibitor of TNF-induced IEC apoptosis in intestinal inflammation. Given that EFHD2 expression was decreased in the intestinal epithelium of UC patients at the active stage (Fig. 1a, c), we next examined whether EFHD2 expression was correlated with UC patients' responsiveness to anti-TNF therapy. We further performed EFHD2 immunostaining of intestinal biopsies from responders and non-responders of UC patients receiving anti-TNF therapy (infliximab treatment). The results showed that the expression level of EFHD2 in the intestinal epithelium of responders was higher than that of non-responders, indicating higher EFHD2 expression might be correlated with better response to anti-TNF therapy in UC patients (Fig. 7g). Overall, the expression pattern and level of EFHD2 in the intestinal epithelium might be predictive of responsiveness to anti-TNF therapy of ulcerative colitis.

Taken together, we hypothesized that in normal IECs, high level of EFHD2 could maintain the low phosphorylation of Cofilin to modulate the dynamic of actin cytoskeleton and block TNFR1 internalization upon TNF stimulation. However, when the level of EFHD2 was much reduced in IECs, TNF engaged TNFR1 and, subsequently, TNFR1 internalized to initiate IEC apoptosis, which drives active inflammation due to the breakdown of epithelium barrier (Fig. 7h).

## Discussion

EFHD2 is a calcium-binding protein with highly variable expression levels and diverse functions in different cell types[15]. Our study defines a suppressive role of EFHD2 in TNF-mediated IEC death as an endogenous inhibitor of TNFR1 internalization, thus providing new insights into the pathogenesis of intestinal inflammation. EFHD2 is recently reported as a cargo-specific adaptor to be involved in the regulation of endocytosis, which directs Rab21-integrins with Arf1, IRSp53, and actin to the clathrin- and dynamin-independent endocytosis. Consequently, EFHD2 supports the migration and invasion of triple-negative breast cancer cells[27]. Furthermore, EFHD2 cross-links actin filaments in a calcium-dependent manner to control the migration of *Drosophila* macrophages[28]. Our findings show that EFHD2 inhibits clathrin-dependent endocytosis of TNFR1 by interacting with the actin-

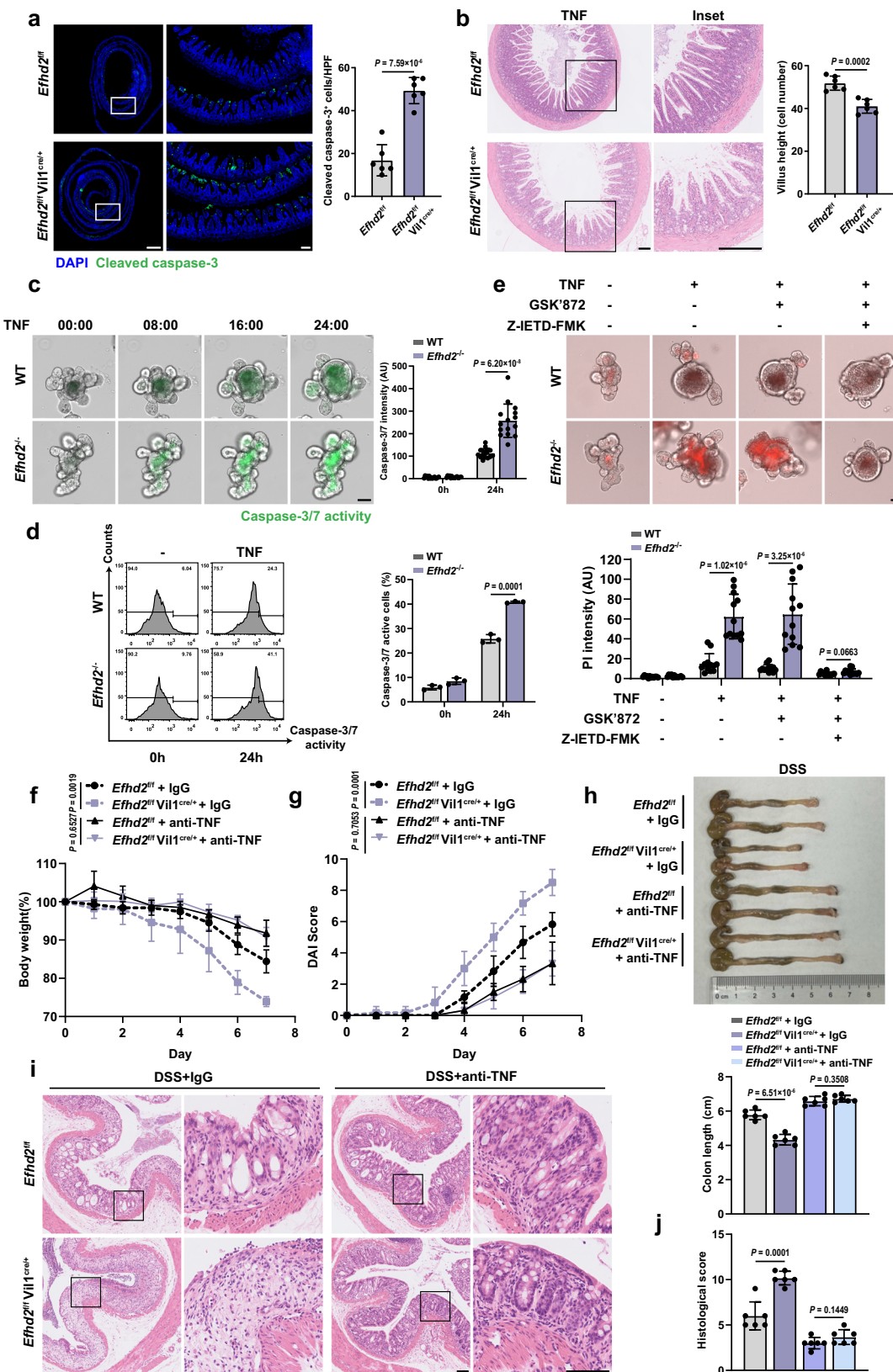

binding protein Cofilin to restrain its phosphorylation, indicating the fundamental biologic function of EFHD2 involved in diverse endocytotic processes and the dynamic of actin filaments.

Excessive production of inflammatory cytokines and cell death factors leads to aberrant immunological and inflammatory immune responses and contributes to the development of autoimmune and inflammatory diseases[29]. Thus, the negative regulation of the inflammatory cytokines for resolution of inflammation attracts much attention now. For example, our previous study showed that the epigenetic modifier Tet2 associates Hdac2 to repress IL-6 during inflammation resolution in myeloid cells, which restrains DSS-induced colitis in mice[30]. Moreover, Rhbdd3 has been reported as a suppressor of the

**Fig. 4 | EFHD2 protects intestinal epithelium from TNF-induced cell death.**
**a** Representative images of immunofluorescence for cleaved caspase-3 (green) and quantification of the small intestine from *Efhd2*^f/f^ and *Efhd2*^f/f^ Vil1^cre/+^ mice 4 h after intravenous injection with TNF. Left scale bar, 1000 μm; right scale bar, 100 μm. *n* = 6 each group. **b** H&E staining of the small intestine from **a** and quantification of villus height. Left scale bar, 100 μm; right scale bar, 200μm. n = 6 each group. **c** Time-series images of WT and *Efhd2*^-/-^ enteroids on day 5 treated with 20 ng/ml TNF in the presence of a caspase-3/7 activity detection reagent (green). Digits represent hours: minutes; scale bars, 50 μm. Quantification of caspase-3/7 intensity was shown in the right histogram. AU arbitrary units. n = 15 enteroids each group. **d** Flow cytometric analysis of caspase-3/7 activity from dispersed WT and *Efhd2*^-/-^ enteroids treated with TNF for 24 hours. *n* = 3 mice each group. **e** Representative images of WT and *Efhd2*^-/-^ enteroids on day 5 treated with 20 ng/ml TNF together with GSK'872 or with Z-IETD-FMK for 24 hours in the presence of propidium iodide (PI) (red). Scale bar, 50 μm. The PI intensity was quantified at the endpoint (below). *n* = 12 enteroids each group. **f**, **g** Relative percentage change of body weight (**f**), DAI score (**g**) of *Efhd2*^f/f^ and *Efhd2*^f/f^ Vil1^cre/+^ mice subjected to 2.5% DSS and injected intraperitoneally with control IgG or anti-TNF antibody. **h** Representative images of colon morphology and lengths from mice in **f** on day 7. **i**, **j** Representative images of H&E staining (**i**) and histologic scores (**j**) of distal colons of mice in **f** on day 7. Upper scale bar, 100 μm; lower scale bar, 200 μm. *n* = 6 per group (**f**–**h**, **j**). Data are representative of at least three independent experiments; error bars show means ± s.d. *P* values were determined by unpaired two-tailed *t*-test. For body weight curves and DAI score, two-way ANOVA analysis with Sidak's multiple comparisons test. Source data are provided as a Source Data file.

production of IL-6 by inhibiting K63-linked polyubiquitination of NEMO in dendritic cells to control Th17 cell-mediated colitis caused by excessive activation of dendritic cells[31]. We once revealed the fundamental role of the receptor membrane translocation of IFN-γR2 in IFN-γ mediated anti-bacterial immune signaling[18]. In IBD pathogenesis, TNF is characterized by inducing excessive IEC apoptosis and necroptosis, which further causes elevated and unresolved intestinal inflammation[4,32]. Thus, identifying the regulators for membrane translocation of inflammatory cytokine or death receptors will deepen the understanding of inflammation and immune disorders. Here we propose a new way of regulating death receptor TNFR1 membrane trafficking, providing mechanistic insight to the functional regulation of death receptors in cell death and inflammatory immune disorders.

Compartmentalization of TNF-R1 signaling is essential to the TNF-induced pro-apoptotic pathway[8]. Although TNFR1 endocytosis is known to be initiated by the formation of clathrin-coated endocytotic vesicles, the inhibitory mechanism of this process remains poorly characterized[9,33,34]. The adenovirus protein 14.7 K can act as an exogenous inhibitor of TNFR1 internalization by blocking the recruitment and formation of the death-inducing signaling complex (DSC). This process does not affect TNF-induced NF-κB activation, implicating an immune escape mechanism by inhibiting host cell death[35]. From another side, we find that EFHD2 inhibits TNFR1 endocytosis to repress apoptotic signaling while promoting NF-κB signaling in IECs. Our findings propose EFHD2 as an endogenous suppressor of TNFR1 internalization in preventing excessive IEC death from driving intestinal inflammation.

TNF-induced cell death signaling is precisely controlled by multi-checkpoints. When cIAPs in complex I are inhibited by a SMAC mimetic such as LCL-161, reduced ubiquitylation of RIPK1 leads to RIPK1-dependent apoptosis. On the other hand, inhibition of the transcription of anti-apoptotic factors in NF-κB signaling by CHX leads to RIPK1-independent apoptosis[36,37]. We show that EFHD2 deficiency can significantly promote TNF-induced IEC apoptosis in both RIPK1-dependent and -independent ways. Moreover, EFHD2 interacts with RIPK1 of Complex I early after TNF engagement with TNFR1, which suggests the role of RIPK1 as a scaffold in the regulation of TNFR1 internalization by EFHD2. However, how the adaptor EFHD2 selectively couples Complex I with Cofilin to modulate clathrin-dependent endocytosis of TNFR1 still needs further study. Meanwhile, we discover that EFHD2 inhibits IECs apoptosis but not necroptosis. Since cytosolic complex II could also initiate necroptosis when caspase-8 is inactive[38], the insignificant effects of EFHD2 on TNF-induced necroptosis might be correlated with the caspase activity or other potential mechanisms that requires further investigation.

Cofilin, as an important actin-binding protein, regulates the assembly and rearrangement of actin filaments, which is widely involved in biological processes like cell migration, adhesion, and division. The precise regulation of Cofilin phosphorylation is crucial for the dynamic of actin cytoskeleton during the invasion process of various pathogens[25,39,40]. Here, we demonstrate the Cofilin's role in the compartmentalization of TNFR1 signaling by showing that EFHD2 interacts with Cofilin to maintain its low phosphorylation level, the active status, for restraining the early endocytosis of TNFR1 complex. In the absence of EFHD2, Cofilin, released from its binding with EFHD2, can be highly phosphorylated to allow the remodeling of actin filaments and facilitate TNFR1 internalization, eventually leading to the enhanced apoptotic signaling. Yet, how cofilin phosphorylation is inhibited by interacting with EFHD2 remains unclear.

It is noteworthy that the EFHD2 expression is significantly decreased in IECs from UC patients and experimental colitis mice at the active stage, compared to the high abundance during the steady state or the remission stage. Thus, under physiological conditions, the high abundance of EFHD2 in IECs protects excessive IEC death by inhibiting TNFR1 internalization, without affecting the total level of TNFR1. When there is a reduction of EFHD2 in IECs, IECs become more sensitive to TNF-induced apoptosis thereby accelerating intestinal inflammation. Therefore, EFHD2 is an important suppressor of IEC death, controlling the pathogenesis and resolution of intestinal inflammation, and the maintenance and restoration of intestinal barrier. Anti-TNF therapy is an effective treatment for ulcerative colitis, while approximately one-third of patients still remain not responsive[41]. Thus, exploring the predictive response signature for anti-TNF therapy will be beneficial to improve the effectiveness of treatment. Besides, it's more evident to assess biomarkers in intestinal mucosa from IBD patients while most studies focus on serum biomarkers[41,42]. Since our study indicated EFHD2's role in inhibiting TNF-induced IEC apoptosis, our histologic analysis on intestinal biopsy samples of UC patients indicated that higher expression level of EFHD2 may be correlated with better response to anti-TNF therapy. Therefore, EFHD2 might be a potential predictor to guide the clinical treatment of ulcerative colitis with anti-TNF therapy.

## Methods
### Ethics statement
All animal experiments conducted in this study were carried out following the guidelines established by the Institutional Animal Care and Use Committee (IACUC) of the Institute of Laboratory Animal Science of Chinese Academy of Medical Sciences. The experimental design and procedures were reviewed and approved by the animal ethics review board (ACUC-A01-2016-006). For human samples, the Ethics Review Board of Peking Union Medical College Hospital approved the study design and experimental procedures, the use of pathological specimens and the review of pertinent patient records (ZS-2178). Informed consent was obtained from all subjects.

### Mice and cell lines
C57BL/6 mice were obtained from Beijing Vital River Laboratory Animal Technology Co., Ltd. (Beijing, China). The *Efhd2* floxed mice (*Efhd2*^f/f^) were bred to induce deletions between the two loxP at exon 2 of the *Efhd2* gene by Bacterial Artificial Chromosomes (BAC)

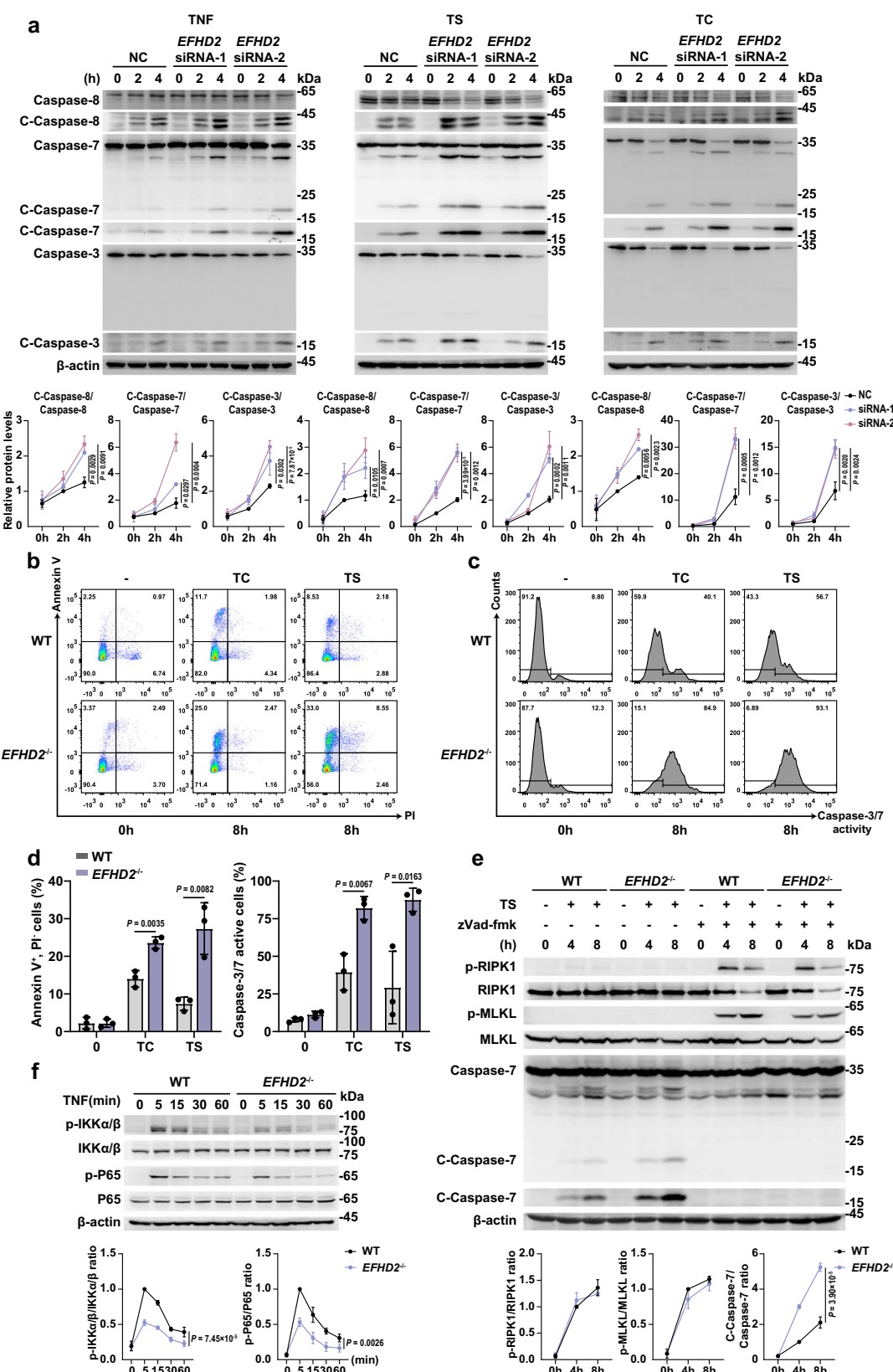

recombineering technology. *Efhd2⁻/⁻* mice were constructed by crossing *Efhd2*^f/f mice with 129 S/Sv-Tg (Prm-cre) 58Og/J mice, *Efhd2*^F/F Lyz^cre/+ mice by crossing *Efhd2*^f/f mice with Lyz-Cre, and *Efhd2*^f/f Vil1^cre/+ mice were constructed by crossing *Efhd2*^f/f mice with Vil1-Cre mice. All mice were genotyped by PCR before experimentation. The genotyping primers were listed in Supplementary Table 1. Both male and female

mice aged 8-10 weeks were used in animal experiments. At experimental endpoints, mice were euthanized by cervical dislocation. All animals were bred in a specific pathogen-free environment, housed with no more than five animals per cage, and kept under controlled lighting conditions (12-hour light/12-hour dark cycle). The temperature was maintained at 24 ± 2 °C, and humidity ranged from 40% to 70%.

**Fig. 5 | EFHD2 suppresses RIPK1-dependent and -independent TNF-induced intestinal cell apoptosis. a** Western blot analysis of lysates from wild-type and *EFHD2* interfered HCT-116 cells treated with 20 ng/ml TNF alone, TNF and Cyclo-heximide (TC) or TNF and LCL-161 (TS) for the indicated time. Corresponding quantifications of cleaved-caspase-8, cleaved-caspase-7, and cleaved-caspase-3 by ImageJ were shown below. *n* = 3. Representative flow cytometric analysis of cell apoptosis (**b**) and caspase-3/7 activity (**c**) from WT and *EFHD2*⁻/⁻ HT-29 cells treated with TC or TS for the indicated time and quantification of proportions of annexin V⁺ PI⁻ apoptotic cells and the percentage of active caspase-3/7 cells (**d**). *n* = 3. **e** Western blot analysis of lysates from WT and *EFHD2*⁻/⁻ HT-29 cells treated with TS and zVAD-fmk for indicated time. Quantifications of p-RIPK1, p-MLKL, and cleaved-caspase-7 were shown below. *n* = 3. **f** Western blot analysis and quantifications of p-IKKα/β and p-P65 in lysates from WT and *EFHD2*⁻/⁻ HCT-116 cells treated with TNF for the indicated time. *n* = 3. Data are representative of three independent experiments; error bars show means ± s.d. *P* values were determined by unpaired two-tailed *t*-test in **d**. Two-way ANOVA analysis with Sidak's multiple comparisons test in **a**, **e**, **f**. Source data are provided as a Source Data file.

Mice were fed a standard laboratory chow (Xietong Bio Inc., XT101FZ-002).

HCT-116 and HT-29 cell lines used in this study were obtained from Cell Resource Center, Peking Union Medical College (PCRC). These cells were cultured in RPMI-1640 medium, which was supplemented with 10% fetal bovine serum (FBS, Gibco). *EFHD2*-deficient HCT-116 and HT-29 cells were generated using CRISPR-Cas9 by utilizing two sgRNAs that targeted exon 1 of the *EFHD2* gene and two sgRNAs that targeted exon 2 of the *EFHD2* gene (Supplementary Table 1).

### Human Specimen Analysis
The human intestinal biopsy samples for this study were obtained from the colons of UC patients at the active or remission stage and healthy controls. They were collected through colonoscopic biopsy by Peking Union Medical College Hospital. The diagnosis of UC was confirmed based on standard clinical judgment. The details of sample information were listed in Supplementary Table 2. The human intestinal colonoscopic biopsy samples from UC patients that were responsive and non-responsive to infliximab treatment were collected by Peking Union Medical College Hospital, and diagnosed based on standard clinical judgment. The relative details of sample information were listed in Supplementary Table 3.

### Plasmids and transfection of cells
The Flag-tagged human EFHD2 ORF clone was purchased from Origene (RC204948). HCT-116 and HT-29 cells were transfected with Lipofectamine 3000 (Thermo Fisher Scientific) following the manufacturer's instructions.

### DSS administration and induction of experimental acute colitis
For acute experimental colitis induction, 8 to 10-week-old mice were given a 2.5% or 3% DSS (MP Biochemical) in their drinking water for 6 days, followed by normal drinking water for the remainder of the experiment until day 7. The body weight, stool consistency, and posture of each mouse were monitored daily to calculate the DAI, which is a combined score of weight loss, stool consistency, and body posture. The DAI was determined as follows: changes in weight loss (0: less than 5%, 1: 5-10%, 2: 10-15%, 3: more than 15%); stool consistency (0: normal, 1: mild loose stools, 2: loose stools, 3: diarrhea and bloody stools); body posture (0: smooth fur without a hunchback, 1: mild fur and a hunchback, 2: moderate fur and a hunchback, 3: severe fur and a heavy hunchback). At the end of the experiment, the mice were euthanized and the intestine tissues were collected for further measurements and analysis. Mice subjected to DSS treatment were injected intraperitoneally with anti-mouse TNF antibody or IgG control (5 mg/kg; BioXCell) every two days, starting from the initiation of DSS administration as previously described[43].

### In vivo administration of recombinant TNF
*Efhd2*^f/f Vil1^cre/+ and *Efhd2*^f/f mice were injected with recombinant murine TNF (0.3 mg/kg, PeproTech) intravenously and sacrificed 4 hours after injection. The small intestine tissues were collected immediately for further histology analysis and immunofluorescence staining.

### Intestinal permeability assay
Mice were orally gavaged with FITC-dextran (Sigma #FD4) at a dosage of 60 mg per 100 g of body weight. Four hours later, plasma was collected and the concentration of FITC was assessed using a SynergyH1 automatic microplate reader (Biotek).

### Isolation of epithelial cells and lamina propria cells
The colons were removed and opened lengthwise, then rinsed thoroughly three times with DPBS. They were then transferred into D-Hank's buffer containing 1 mM DTT and incubated by shaking for 10 minutes at 37 °C. The supernatants were discarded and replaced with fresh D-Hank's buffer containing 2 mM EDTA and incubated for an additional 20 minutes at 37 °C by shaking. The supernatants were collected, and the intestine was washed three times with DPBS. All supernatants were passed through a 100-μm cell strainer to obtain single-cell suspensions, which were collected as the IEC fraction. The remaining lamina propria was then cut into 2 mm sections and digested for 45 minutes at 37 °C using a digestive solution consisting of DMEM supplemented with 10% FBS, 0.5 mg/ml collagenase IV, 3 mg/ml dispase II, and 0.5 mg/ml DNase I. A single-cell suspension was obtained by sequentially passing the sample through a 70-μm cell strainer followed by a 40-μm cell strainer. The suspension was centrifuged and then subjected to Percoll gradient isolation, with the interface cells being collected as lamina propria cells. Further mass cytometry by time-of-flight (CyTOF) was performed according to an established protocol[44], and then analyzed using Halios (Fluidigm). The antibodies used during the CyTOF analysis were all purchased from Fluidigm. All mass cytometry files were analyzed using Cytobank software and R cytofkit package[45].

### Mouse intestinal epithelial cell organoid culture
The culture of mouse intestinal epithelial cell organoids was performed according to an established protocol[46]. Briefly, a 15 cm segment of the small intestine was harvested from a mouse, longitudinally cut, and gently washed with cold DPBS. The segment was then divided into 5 mm pieces and digested in a solution of cold DPBS supplemented with collagenase I for 15–20 minutes at 37 °C in an incubator. The segments were then rinsed with sterile DPBS and subjected to repeated pipetting to mechanically isolate the crypt cells. The suspension was centrifuged to enrich intestinal crypts. The purity and density of the crypts were assessed under an inverted microscope. The crypts were suspended in a mixture of an equal volume of culture medium, IntestiCult Organoid Growth Medium (STEMCELL), and Matrigel matrix (Corning), and seeded at a density of 200 crypts per well for culture. The plate was then incubated at 37 °C for 30 minutes till the Matrigel matrix solidified, and then 500 μl of culture medium was added, exchanged every 3 days.

### Cell death assay and flow cytometry analysis
Induction of apoptosis was performed using a combination of 20 ng/ml recombinant mouse or human TNF (PeproTech), 10 μg/ml CHX (Selleckchem), and 5 μM LCL-161 (MedChemExpress) as reported[23]. The concentrations of GSK'872 (Selleckchem) and Z-IETD-FMk (Selleckchem) used were 10 μM as reported[23]. To induce

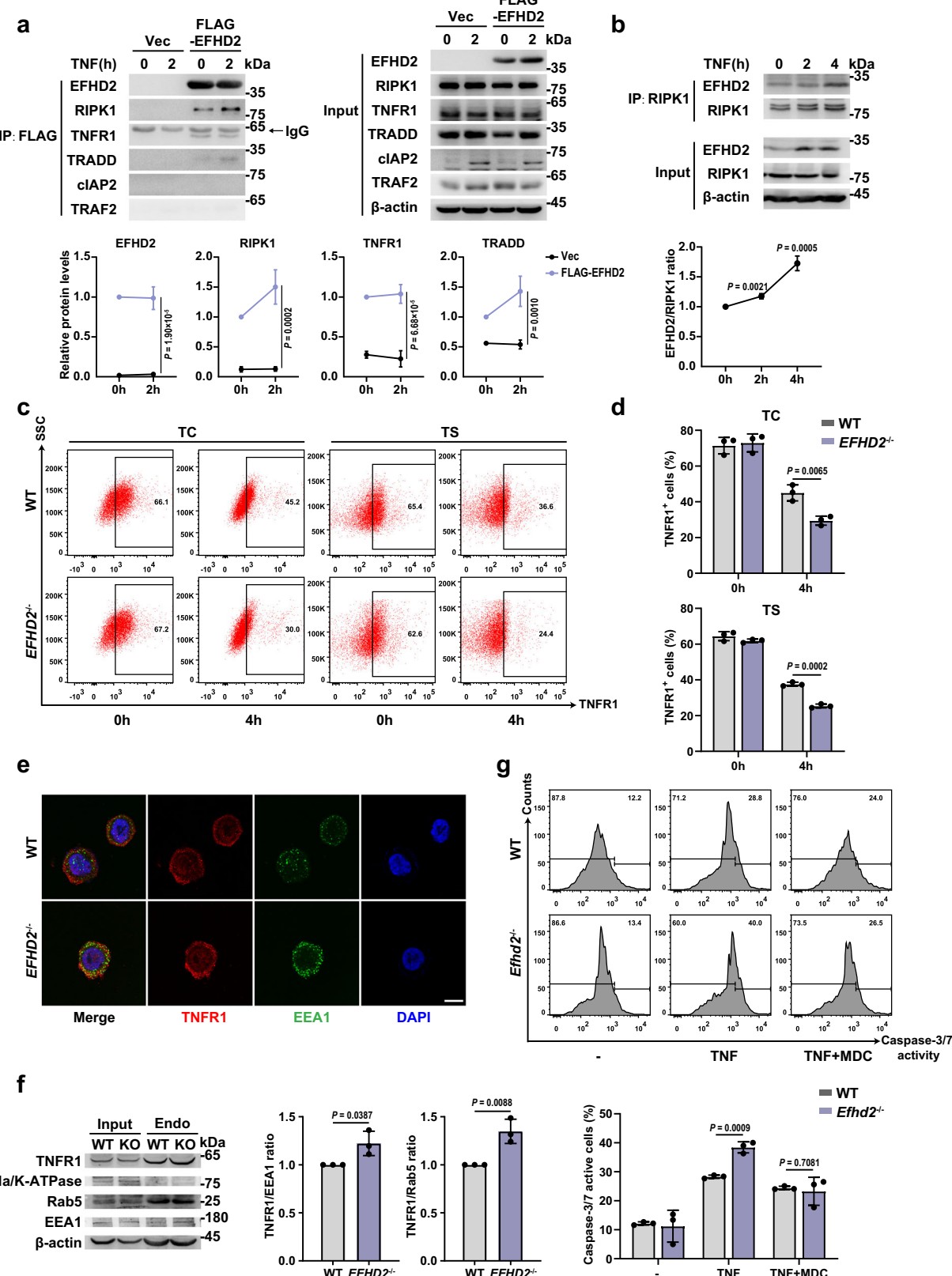

necroptosis, the same concentrations were used, with the addition of 20 μM zVAD-fmk (Selleckchem) as reported[23]. At the endpoint of the experiment, cells were stained with the FITC Annexin V Apoptosis Detection Kit (BD Biosciences) following the manufacturer's instructions and analyzed using a BD LSRFortessa flow cytometer (BD Biosciences). To detect the cell-surface level of TNFR1, cells were

stained with APC-CD120a (Biolegend). Flow cytometry data were analyzed by Flowjo_V10. For live imaging of organoid cell death, the organoids were incubated with CellEvent Caspase-3/7 Green Detection Reagent (Thermo Fisher Scientific) and PI (BD Biosciences) for 30 minutes at 37 °C. Images were acquired using the ZEISS Celldiscoverer 7 imaging system (ZEISS).

**Fig. 6 | EFHD2 interacts with complex I and inhibits the internalization of TNFR1. a** Lysates from HCT-116 cells without or with overexpression of Flag-EFHD2 treated with 50 ng/ml TNF for the indicated time were immunoprecipitated with Flag M2 beads followed by western blot analysis for indicated molecules of TNFR1-complex I. Quantifications of the indicated proteins in TNFR1-complex I were shown below. **b** Lysates from HCT-116 cells treated with TNF for indicated time were immunoprecipitated with RIPK1 to analyze EFHD2 interaction by western blot. The quantification of EFHD2/RIPK1 ratio was shown below. Representative flow cytometric analysis plot (**c**) and proportions (**d**) of cell-surface TNFR1-positive cells from WT and *EFHD2*$^{/-}$ HT-29 cells treated with TC, or TS for the indicated time. **e** Representative immunofluorescence detection of TNFR1 (red) and EEA1 (green)

from WT and *EFHD2*$^{/-}$ HT-29 cells treated with TS for 30 min. Scale bar, 10 µm. **f** Western blot detection of lysates from isolated early endosome from WT and *EFHD2*$^{/-}$ HT-29 cells treated with TS for 30 min (left) and quantifications of TNFR1/EEA1 and TNFR1/Rab5 ratio in early endosome by ImageJ (right). **g** Flow cytometric analysis of caspase-3/7 activity from dispersed WT and *Efhd2*$^{/-}$ enteroids treated with 20 ng/ml TNF for 24 hours, without or with MDC pretreatment for 1 hour. *n* = 3 (**a**, **b**, **d**, **g**). Data are representative of three independent experiments; error bars show means ± s.d. *P* values were determined by unpaired two-tailed *t*-test in **b**, **d**, **f**, **g**. Two-way ANOVA analysis with Sidak's multiple comparisons test in **a**. Source data are provided as a Source Data file.

## Histological and immunohistological analyses

Briefly, intestine sections with a thickness of 5-10 µm were fixed overnight in 4% paraformaldehyde and processed for paraffin embedding. H&E staining was performed by staining the sections with hematoxylin and dehydration in graded alcohols and xylene. For PAS staining, the dehydrated paraffin sections were stained with periodic acid, incubated with Schiff's reagent, and then counterstained with hematoxylin before dehydration. Histologic scoring of epithelial damage for colon tissues was determined as follows: the crypt architecture was rated on a scale from normal (0) to severe crypt distortion with loss of entire crypts (3); the degree of inflammatory cell infiltration was rated on a scale from normal (0) to dense inflammatory infiltrate (3); muscle thickening was rated on a scale from normal (0) to marked muscle thickening (3); Goblet cell depletion was rated on a scale from absent (0) to present (1); and crypt abscess was rated on a scale from absent (0) to present (1). For the immunohistochemistry experiment, the sections were stained with indicated primary antibodies: anti-EFHD2 (1:800, Sigma Aldrich, HPA048961); anti-Efhd2 (1:800, Abcam, ab24368); anti-Chromagranin-A (1:250, Abcam, ab15160); anti-Lysozyme (1:1000, Abcam, ab108508); anti-Muc-2 (1:500, Servicebio, GB11344-100); anti-Ki67 (1:600, Servicebio, GB121141-100); anti-CA1 (1:2000, Abcam, ab267475). Images were acquired using Aperio XT (LEICA) and NanoZoomer S360 (Hamamastu).

## Immunofluorescence staining

HCT-116 and HT-29 cells were fixed in 4% polyformaldehyde, blocked with 10% donkey serum, and permeabilized in 0.1% Triton X-100/PBS (Sigma Aldrich). To detect endosomal localization of TNFR1, the cells were then incubated with anti-TNFR1 (1:400, Santa Cruz, sc-8436) and anti-EEA1 (1:200, Cell Signaling Technology (CST), #3288) primary antibodies overnight at 4 °C, followed by incubation with fluorophore-labeled secondary antibodies (1:500, Thermo Fisher Scientific, A-11005, A-11008) and DAPI (1:1000, CST). For paraffin sections of intestinal tissues, specific primary antibodies were used, including anti-cleaved-caspase-3 (1:300, CST, #9661), anti-cleaved-caspase-8 (1:100, CST, #8592), anti-p-MLKL (1:200, CST, #37333), and anti-E-cadherin antibody (1:500, Servicebio, GB12082-100). TUNEL staining was performed with the TUNEL BrightGreen Apoptosis Detection Kit (Vazyme). Fluorescence images were acquired using Leica TCS SP8 gSTED 3X (Leica) and Pannoramic MIDI (3DHISTECH).

## Immunoblot and immunoprecipitation

Cells were lysed with cell lysis buffer (CST) or IP lysis buffer (Thermo Fisher Scientific). Immunoblot and immunoprecipitation assays were then performed as previously described[30]. For immunoblot analysis, the following antibodies were used: anti-caspase-7 (1:1000, CST, #9492), anti-caspase-3 (1:1000, CST, #9662), anti-caspase-8 (1:2000, CST, #9746), anti-cleaved-caspase-8 (1:1000, CST, #9496, #8592), anti-p-MLKL (1:1000, CST, #91689), anti-MLKL (1:1000, CST, #14993, #37705), anti-p-RIPK1 (1:1000, CST, #65746, #31122), anti-RIPK1 (1:1000, CST,

#3493), anti-p-IKKα/β (1:2000, CST, #2697), anti-IKKα (1:1000, CST, #2682), anti-IKKβ (1:1000, CST, #8943), anti-p-P65 (1:2000, CST, #3033), anti-P65 (1:1000, CST, #8242), anti-TRADD (1:1000, CST, #3684), anti-cIAP1 (1:1000, CST, #7065), anti-cIAP2 (1:1000, CST, #3130), anti-TRAF2 (1:1000, CST, #4724), anti-Rab5 (1:1000, CST, #3547), anti-EEA1 (1:1000, CST, #3288), anti-p-Cofilin (1:1000, CST, #3313), and anti-Cofilin (1:1000, CST, #5175), anti-Na/K-ATPase (1:2000, Santa Cruz, sc-48345), anti-β-actin (1:5000, MBL, M177-3), anti-EFHD2 (1:1000, Genetex, GTX108080), anti-Flag (1:5000, Sigma Aldrich, F1804), anti-TNFR1 (1:600, Proteintech, 21574-1-AP), anti-cFLIP (1:1000, CST, #56343), anti-Bcl-xl (1:1000, CST, #2764), and anti-Bcl-2 (1:1000, CST, #3498). Anti-FLAG M2 Magnetic Beads (Sigma Aldrich) and RIPK1 (1:100, CST, #3493) were used for immunoprecipitation assays in HCT-116 cells. Proteins that were immunoprecipitated by FLAG beads were separated and stained by silver, followed by liquid chromatography-tandem mass spectrometry (LC-MS). The resulting peak lists were used to search the NCBI human protein database with the Mascot search engine. The candidates that interacted with EFHD2 were listed in Supplementary Table 4. Endosomes of HT-29 cells (approximately 3-5 × 10$^7$ cells) were isolated using the Endosome Isolation and Cell Fractionation Kit (Invent) according to the manufacturer's instructions. Quantifications of immunoblots were analyzed by ImageJ (V 1.8.0).

## Quantitative real-time PCR (qPCR) assay

Extraction of total RNA from cells and animal tissues was carried out using TRIzol reagent (Thermo Fisher Scientific) and RNAfast200 kit (Fastagen) according to the manufacturer's instructions. 1 µg of RNA was reverse transcribed, and 40 ng of cDNA was further used for the Real-time PCR assay. Real-time PCR analysis was performed by QuantStudio 7 Flex Real-Time PCR System (Thermo Fisher Scientific) using RealStar Power SYBR qPCR Mix (GenStar). Sequences of the primers for qPCR were listed in Supplementary Table 5.

## RNA interference

EFHD2-specific small interfering RNAs (siRNAs) were procured from RiboBio. The RNA interference was carried out in HCT-116 or HT-29 cells using RNAi MAX reagent (Invitrogen), following the manufacturer's protocol. The efficacy of the interference was then evaluated by qPCR and immunoblot assay. Sequences of siRNAs were listed in Supplementary Table 5.

## Transcriptome analysis

Primary IECs were isolated from colons of *Efhd2*$^{f/f}$ Vil1$^{cre/+}$ and *Efhd2*$^{f/f}$ mice, with and without DSS administration. Total RNA was then extracted using TRIzol reagent. After confirming the quantity and quality of RNA, the library was constructed using the NEBNext NltraTM RNA library Prep Kit for Illumina (NEB) according to the manufacturer's instructions. Transcriptome sequencing was performed using the Illumina NovaSeq 6000 platform (Novogene). Differential expression analysis between the indicated two groups was analyzed by the DESeq2 R package. |Log$_2$Fold Change|> 1 and padj <0.05 were set as the

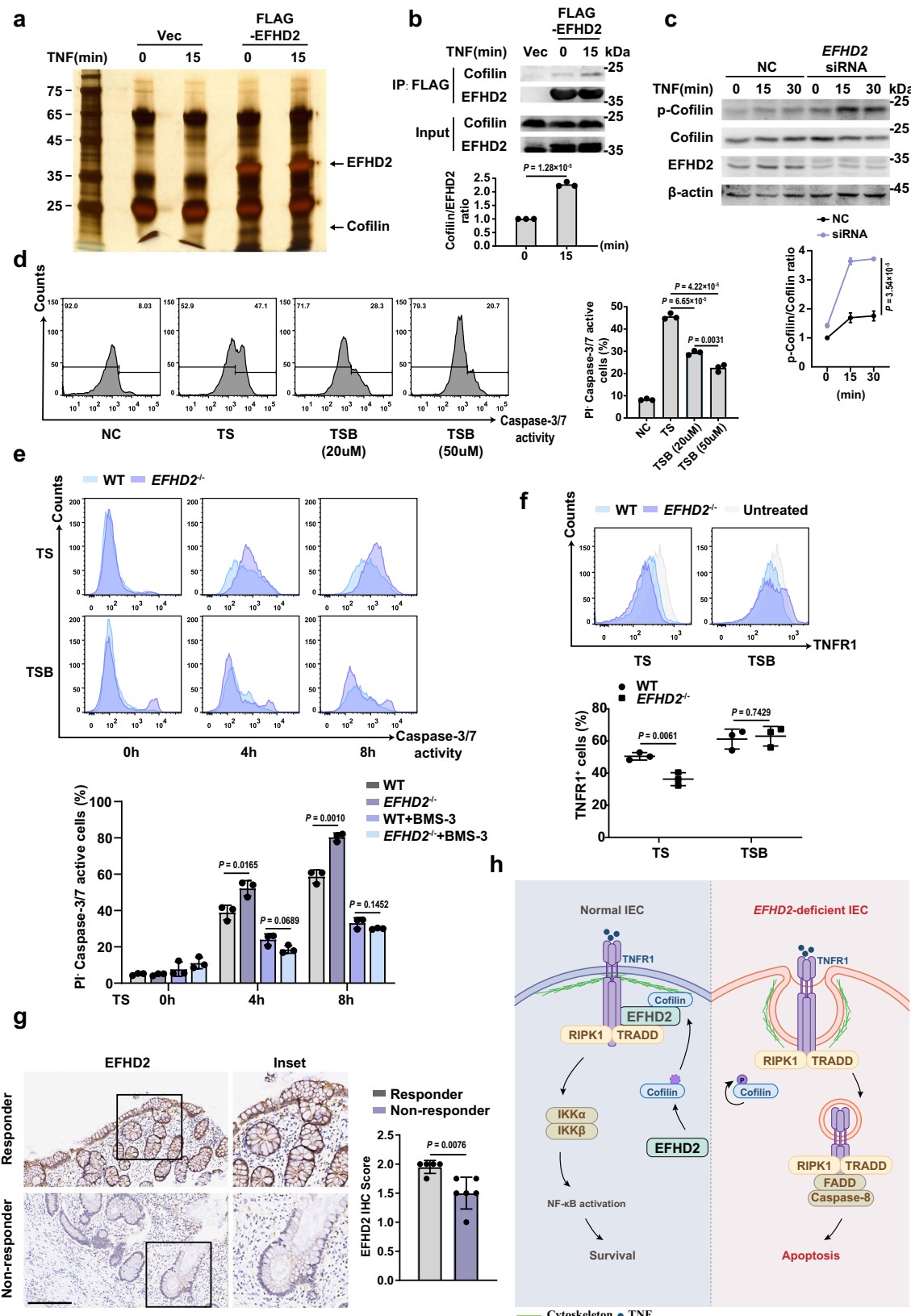

threshold for significantly differential expression. GO and KEGG enrichment analysis of differentially expressed genes were implemented by the clusterProfiler R package, with a threshold of padj <0.05 set for significant enrichment. The sequencing data are deposited in the NCBI GEO dataset under accession code GSE224768.

## Statistical analysis

Statistical analysis was performed using the GraphPad Prism 8.0. For comparisons, two-tailed Student's *t*-test and two-way ANOVA analysis with Sidak's multiple comparisons test were performed. Data are shown as means ± s.d. *P* < 0.05 was considered as significant.

**Fig. 7 | EFHD2 interacts with Cofilin and inhibits Cofilin phosphorylation to block TNFR1 internalization. a** Silver staining of the gel separating proteins immunoprecipitated by FLAG beads from lysates prepared from HCT-116 cells transfected with FLAG-EFHD2 treated with 50 ng/ml TNF for the indicated time. **b** Co-IP analysis of the interaction between FLAG-EFHD2 and Cofilin in HCT-116 cells treated with TNF for the indicated time by western blot. The quantification of Cofilin/EFHD2 ratio by ImageJ was shown below. **c** Western blot analysis and quantification of the p-Cofilin in wild-type and *EFHD2* interfered HCT-116 cells treated with TNF for the indicated time. **d** Flow cytometric analysis (left) and proportions (right) of PI⁻ caspase-3/7 active cells from HCT-116 cells treated with TS for 4 hours, without or with 20 μM or 50 μM BMS-3 pretreatment (TSB). **e** Flow cytometric analysis and proportions of PI⁻ caspase-3/7 active cells from WT and *EFHD2⁻/⁻* HT-29 cells treated with TS for the indicated time, without or with 20 μM

BMS-3 pretreatment. **f** Flow cytometric analysis and proportions of cell-surface TNFR1-positive cells from WT and *EFHD2⁻/⁻* HT-29 cells treated with TS for 30 minutes, without or with 20 μM BMS-3 pretreatment. *n* = 3 (**b**–**f**). **g** Representative images of immunostaining for EFHD2 on intestinal biopsy samples from UC patients that were responsive (*n* = 5 patients) and non-responsive (*n* = 6 patients) to infliximab treatment. Histologic scores of EFHD2 staining were using a 3-point quantification scale. Scale bar, 300 μm. **h** Model depicting how deficiency of EFHD2 promotes TNFR1 internalization and intestinal inflammation due to excessive apoptosis. Data are representative of three independent experiments; error bars show means ± s.d. *P* values were determined by unpaired two-tailed *t*-test in **b**, **d**–**g**. Two-way ANOVA analysis with Sidak's multiple comparisons test in **c**. Source data are provided as a Source Data file.

## Reporting summary

Further information on research design is available in the Nature Portfolio Reporting Summary linked to this article.

## Data availability

The RNA high throughput sequencing data of this study is deposited in the NCBI GEO dataset under accession code GSE224768. The public single-cell RNA sequencing dataset in the re-analysis for *EFHD2* expression levels in multiple cell types from normal human colon tissue is deposited in the NCBI GEO dataset under accession code GSE116222. The mass spectrometry data reported in this study is deposited in the ProteomeXchange with the dataset identifier PXD048587. All other study data are included in the article and/or Supplementary Information. Any additional information is available upon request to the corresponding author (Xuetao Cao, caoxt@immunol.org). Source data are provided with this paper.

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

## Acknowledgements

We thank Dr. J. Yang, Dr. J. Wang and Dr. Y. Pu (Peking Union Medical College) for helpful suggestions. This work was supported by National Natural Science Foundation of China (82388201 to X.C.), CAMS Inno-vation Fund for Medical Sciences (2021-I2M-1-017 to X.C.) and National Key Research and Development Program of China (2021YFA1201102 to X.X.).

## Author contributions

X.C. supervised the study; J.W., J.D. and Y.C. performed mouse experi-ments in vivo; J.W. X.X., and J.D. performed cell death assays in vitro and signaling experiments; J.W., X.X., Y.H. and J.X. generated and identified the knock-out mouse; J.W., X.X., J.S., D.G., and Y.D. performed the immunofluorescence and flow cytometry; Y.L. provided reagents; T.H and J.L. provided clinical samples and helpful suggestions; J.W., X.X., B.W., and X.C analyzed data and wrote the paper.

## Competing interests

The authors declare no competing interests.
