## [Peer Review File · Nature Communications]

EFHD2 Suppresses Intestinal Inflammation by Blocking Intestinal Epithelial Cell TNFR1 Internalization and Cell DeathREVIEWER COMMENTS

Reviewer #2 (Remarks to the Author):

The study looks at the mechanism by which Efh2 suppresses inflammation in intestinal epithelial cells by blocking phosphorylation of Cofilin and inhibiting internalization of TNFR1 to mediate RIPK1-induced apoptosis. The role of TNFR1 in driving RIPK1-mediated apoptosis via activation of caspases in the intestinal epithelium is well-described in the literature. The authors have extended this field by demonstrating a role for Efh2 in inhibiting phosphorylation of Cofilin in this pathway. While the *in vivo* experiments are comprehensive, and the overall methodology is well-designed, there are a number of issues that influence the impact and support for the mechanisms proposed, which are outlined below. There is also a need for improvement in the scientific language throughout the paper, citations where indicated, a quantitative analysis of all the immunoblots, and information on the human patient data. Histology and immunostaining images lack representation of controls, and secondary staining controls, that need to be added to the body of the manuscript or to supplementary data. Overall, the study is sound and contributes to the advancement of the field of intestinal epithelial cell death signalling.

Major Comments:

Comments on figures:

Figure 1a

- Details on Control and UC patients are missing. The authors should consider including the median age, gender, medical history if available and region of the colon where the tissue was obtained from.
- A quantification of Efh2 should be included for the immunohistochemistry images from control and UC patients to determine if there were any differences in the staining intensity or patterns of Efh2 localization across the patient cohort.
- While the paper is based on a cell death signalling pathway in an *in vivo* colitis model, the authors do not include any discussion pertaining to IBD models or the use of anti-TNF therapy in UC patients in the discussion section. This needs to be addressed.
- Figure 1a will also benefit from a subset image showing a magnified view of Efh2 localization to epithelial cells (and immune cells, too?). A secondary control for the antibody should be included in the supplementary data, as the staining appears quite ubiquitous here.
- Did the authors see any decrease in TNFR1 or caspases corresponding with decreased Efh2 in the UC tissues compared to those in remission or in controls?

Figure 1b

- Instead of a n=2 representation, this figure would benefit from a subset image in the lower panel showing a magnified view of Efh2 localization to epithelial cells and immune cells with the secondary controls.
- Does Efh2 localize in a similar manner or to the same levels of expression in different regions of the mouse and human intestine? This is significant, since the authors have focussed on small intestine, enteroids, or the colon for different parts of the study.

Figure 1f

- Figure labels indicating the timepoint in the DSS experiment are missing in this panel. The figure will benefit from showing a D0 control, instead of a n=2 representation and the inclusion of a lower magnification of the image that shows the extent of damage in DSS in the whole intestine.
- Please include the same for Figure 2d and e.

Figure 2

- Fig. 2h, line 142-145, the authors mention an "obvious decrease of IECs and impaired intestinal barrier upon DSS treatment" in Efh2f/fVil1cre/+ mice compared to Efh2f/f controls due to loss of E-cadherin. There is no quantitative measurement of barrier function in these mice, so the conclusion needs to be more circumspect or a functional measurement of the barrier should be included.

Figure 3

- A magnified subset image showing the specific localization of Casp3, 8 and p-MLKL should be included in Fig.3b.

- Considering the ubiquitous expression of *Efhd2* on epithelial cells, is the protection conferred limited only to surface epithelial cells or throughout the crypt axis? Fig.3b shows that deletion of *Efhd2* in the epithelium results in significant cell death only in the surface epithelium. Can the authors include a comment on it or further analysis?
- The immunoblot showing p-MLKL and cleaved caspase 3 activity should be quantified and analysed statistically to show the differences in the N numbers.
- Fig. 3e, does not mention the timepoint of the DSS experiment used for Bulk RNA-Seq analysis. This should be clearly indicated in the Figure legend.

Figure 4

- Were the doses of TNF, Cycloheximide and LCL161 determined from a dose-curve in vitro or in vivo or as a guideline from literature? Rationale needs to be included for the choices.
- Did the authors see significant blunting of villi and crypts in *Efhd2f/fVil1cre/+* animals compared to *Efhd2f/f* mice in Fig. 4b? This panel will benefit from the inclusion of a low magnification image and a quantification of the blunted villi and crypts in the small intestine. Did the authors see any changes in the crypt length of the colon of *Efhd2f/fVil1cre/+* mice?
- Is there a reason the authors did not include the TNF only controls for the enteroids experiments in Fig. 4c and d? At what point during organoid growth were TC or TS added to the cultures? The enteroids at T0 appear to be immature (little to no budding) and significantly different in terms of size for WT vs *Efhd2-/-*.
- Although the *Efhd2-/-* organoids +TC/TS appear to be highly apoptotic, they also seem much bigger than controls. Have the authors determined if *Efhd2* influences proliferation, or its loss causes an increase in proliferation and budding in organoids or in the mouse native tissue?
- Figure 5 and 6 requires the immunoblots to be quantified with statistics.
- Supplementary Figures: It would be beneficial to see immunohistochemistry staining for *Efhd2* in the *Vil1Cre/+* and *Efhd2-/-* mice, relative to its controls.
- Were there any inherent differences in the development and differentiation of epithelial lineages in either the enteroids or the native epithelium of *Efhd2f/fVil1cre/+* mice compared to *Efhd2f/f* controls?

General minor comments on the paper

- The conclusion statement in the abstract needs to be re-written to align with the data presented.
- Line 41, IEC- define abbreviations when used for the first time.
- Line 55, TNFR1 cell death signalling should be corrected to TNFR1-mediated cell death signalling.
- Line 56, the spelling of "translocation" should be corrected.
- The formatting of the mouse genotypes in the manuscript should be consistent.
- Line 133, there is no quantification to reflect the differences that justify changes in intestinal development and differentiation in Supplementary Fig. 3i.
- Line 156, 228, 303, 335- include a citation.
- Line 161, include a citation for p-MLKL as a classical marker of necroptosis.
- Several sentences require revision for clarity, see for example- Line 171,179, 187, 195, 197, 222, 363, 368 and 380-385.
- Line 211-212, both cells should be corrected to both "cell lines".
- Line 222, "EFHD2-deficiency cells" should be corrected to "EFHD2-deficient cells".
- Line 307-311 is a complicated sentence – it should be revised for clarity and precision.
- Line 321, rewrite "we report that..." since it is not a part of this study, it was a previous study done by the group.
- Line 329, "cause" should be corrected to "causes".
- Line 456, a single cell suspension cannot be obtained through a 70µm filter, it requires a 40µm filter.
- If the concentrations of CHX and LCL161 were determined from literature, cite references.

Reviewer #3 (Remarks to the Author):

In their manuscript "EFHD2 Suppresses Intestinal Inflammation by Blocking Intestinal Epithelial Cell

TNFR1 Internalization" Wu et al. describe a role for EFHD2 as a negative regulator of IEC death upon TNF-stimulation. EFHD2 deficiency, as it is found in UC, leads to aberrant IEC death and associated intestinal inflammation while its presence can protect IECs from TNF-induced apoptosis.

Major comments

While several dysregulated pathways are responsible for the aberrant and continuous inflammation in IBD, the findings of this paper describe an interesting further mechanism that drives inflammation in UC. Overall, the paper is well written and the experiments performed are appropriate.

The authors use organoid cultures to examine TNF-induced RIPK1-dependent and -independent apoptosis. However, the functionality and viability of the differentiated enteroids need to be confirmed in EFHD2^{-/-} as well as wt enteroids. Please perform functional assays as well as stainings for enteroid markers. Also, please perform stainings for necrotic cores to exclude their formation within the enteroids. In addition to confirming the functionality and viability of enteroids, this would also help to exclude any differences in the development and functionality between EFHD2^{-/-} and wt enteroids.

Fig. 5: Why was TNF stimulation performed in HCT116 cells and TNF+CHX and TNF+LCL-161 stimulation as well as necroptosis-induction in HT29 cells? If all experiments are performed in both cell lines, for consistency please show results for one cell line in the main figures and the results for the other cell line in the supplementary section. This also applies to Fig. 6. In Fig. 5a caspase 3 expression is very inconsistent over the time measured and between wt, siRNA1 and siRNA2. (b) Caspase 8 cleavage was measured in the HT29 stimulation experiments but not in the HCT116 experiments; please explain why. Also, please label the kDa of the proteins cleaved and uncleaved in all western blot pictures.

Fig. 5g: The result section states that under TNF treatment phosphorylation of IKK α /b and p65 in EFHD2^{-/-} cells was significantly decreased compared to wt. However, Fig. 5g doesn't show any difference in IKK α /b phosphorylation between EFHD2^{-/-} cells and wt. Also, IKK α /b expression seems inconsistent over the time it was measured. For p65, phosphorylated p65 is present in EFHD2^{-/-} cells as well as wt cells but then phosphorylated p65 decreases faster in EFHD2^{-/-} cells compared to wt. Is there a reason why phosphorylated p65 is initially also present in the EFHD2^{-/-} cells and/or why it decreases faster in the EFHD2^{-/-} cells? Quantification of the proteins of interest could help to support the statement.

Line 258/Fig. 6f: The main text mentions that there were significantly higher levels of TNFR1 in early endosomes in EFHD2^{-/-} HT29 cells compared to wt cells. However, the graph in Fig. 6f doesn't show any significance. Please also perform an n of 3.

Minor comments

Considering that EFHD2 has previously been shown to be upregulated in various cancer tissues and has been associated with metastasis, the therapeutic potential of EFHD2 must be taken with caution. This is especially the case as the authors found EFHD2 to be only downregulated during active UC while its expression recovered during remission.

Line 172: Please change to "It revealed that genes with an increased expression in EFHD2^{-/-} IECs compared to controls were associated with the acute inflammatory response..."

Line 545: From which tissue were the endosomes isolated and how much tissue was used for endosome isolation?

qPCR: How much RNA was used for RT-PCR/qPCR?

Figures:

Please label the kDa of the proteins shown on the Western Blots, especially when several bands are visible. What are the other bands that are visible?

Fig. 1b: The legend mentions that mice were treated for the indicated times, however, mice were treated for 7d and then left without treatment until d12.

Fig. 1f+g: Please mention after how many days of DSS treatment the staining was performed.

Fig. 2g: This should read "Relative gene expression of *Efhd2*^{f/f} and *Efhd2*^{f/f} 737 *Vil1cre*⁺ mice after 7d of DSS treatment compared to d0".

Fig. 3b: For completion, please add a zoomed-in picture of p-MLKL as it was done for the other conditions.

Fig. 3d: In Fig. 3b + c Caspase 3, caspase 8 and p-MLKL have been analysed, therefore, the western blot in Fig. 3d should also show Caspase 8.

Fig 4: Fig. 4 states that data are representative of at least 2 independent experiments. However, a minimum of 3 independent experiments should be performed.

Fig 4c+d show that both treatments, TNF+CHX and TNF+LCL-161, induce an increase of Caspase3/7 in *EFHD2*^{-/-} enteroids compared to wt. However, in Fig. 4e only TNF+LCL-161 was further analysed; why was TNF+CHX not analysed any further?

Fig. 6 c+d: Please perform a minimum of n=3. Please explain in one sentence why TS was also measured after 0.5 h; is TNFR1 quicker internalised after TS treatment compared to TC treatment? If so, Fig. 6d should be a Suppl. Fig but also include TC treatment after 0.5h.

Fig. 6g: Why was this experiment performed with TNF+LCL161? Would it be sufficient if the experiment was performed with TNF alone?

Fig. 7g: (Suggestion) An option to present the right-hand side of the graph where the situation in *EFHD2*⁻ cells is described, would be to cross out *EFHD2*. Instead of drawing an arrow from *EFHD2* to cofilin, an arrow around cofilin itself could be drawn to show its self-phosphorylation.

Suppl. Figures

Suppl. Fig. 1: Comparison of protein expression is difficult if the housekeeper gene is expressed differently. Has the same amount of protein been loaded onto each lane?

Suppl. Fig. 2: Please perform a minimum n of 3.

Suppl. Fig 2b: From which tissue was the RNA isolated that was used to measure *efhd2* expression in the created K/O mice?

Suppl. Fig 2e+f: To strengthen the data, please perform a minimum of n=3.

Suppl Fig. 5b+c: The main text states that *EFHD2* was knocked down in HCT 116 and HT29 cells, however, the figures only show HCT116 cells.

Suppl. Fig 5c: What is 0, T and TZ? Please explain in the legend. Why does the amount of EFHD2 protein in NC increase from 0 over T to TZ? Was the relative gene expression in Fig. 5b measured at 0, T or TZ?

Reviewer #4 (Remarks to the Author):

Synopsis:

In their paper entitled "EFHD2 Suppresses Intestinal Inflammation by Blocking Intestinal Epithelial Cell TNFR1 Internalization", the authors use a variety of approaches to study how EFHD2 restrains intestinal epithelial cell (IEC) apoptosis by inhibiting TNFR1 internalization.

They present IHC of EFHD2 to demonstrate decreased expression in active UC and in DSS colitis. *Efhd2*^{-/-} mice exhibited increased sensitivity to the DSS colitis model. Conditional deletion of *Efhd2* in IECs, but not myeloid cells, increased sensitivity to DSS colitis. *Efhd2* IEC-deficient mice treated with DSS exhibited increased cleaved caspase-3 (CC3) and CC8, but not p-MLKL, consistent with increased sensitivity to apoptosis but not necroptosis in vivo. Exogenous TNF also increased CC3 staining in *Efhd2*^{f/f} *Vil1cre*^{+/+} mice. *Efhd2*-deficient enteroids exhibited increased caspase-3/7 activity upon treatment with TNF and CHX or SMAC mimetic, but not under necroptotic conditions using zVAD-fmk. Knock-down or knock-out of EFHD2 in human colorectal cancer cell lines exhibited similar responses to apoptotic stimuli, and immunoblots showed increased CC3 and CC7 but not pRIPK1 or pMLKL in response to TNF with CHX or SMAC mimetic. They then show decreased pIKKα/B and P65 in EFHD2^{-/-} cell lines treated with TNF, suggesting reduced NFκB signaling. EFHD2 co-immunoprecipitated with RIPK1, TNFR1, and TRADD, and EFHD2^{-/-} cell lines exhibited reduced surface and increased endosomal TNFR1. Finally, the authors show that EFHD2 binds to and inhibits phosphorylation of Cofilin, blocking TNFR1 internalization, and this can be reversed with an LIMK inhibitor.

There are multiple strengths to this study, including use of conditional villin-Cre mice to study *Efhd2* in IECs, and primary intestinal enteroids. They also present mechanistic studies on how EFHD2 restricts IEC apoptosis via Cofilin and TNFR1 internalization. However, there are multiple major and minor concerns that would limit the impact of the manuscript. The two most significant are a weak link of relevance to IBD and lack of in vivo rescue.

MAJOR

1. The data presented are insufficient to support the claim that EFHD2 is significantly lower in UC patients. There are a few representative IHC images presented, but there is no quantitation of the 10 patients per group. Importantly, at the mRNA level, EFHD2 exhibits increased overall expression in mucosal biopsies from patients with UC (<https://premedibd.com/genes.html>). If EFHD2 protein expression is discordant with the gene expression data, those data should be presented. It's not clear this would be a good target in IBD, since it restrains apoptosis through scaffolding effects, so it would be difficult to improve its function. It is also not associated with IBD through GWAS. The link between EFHD2 and UC therefore seems tenuous.
2. While the authors present evidence that *Efhd2*^{f/f} *Vil1cre*^{+/+} mice exhibit increased sensitivity to IEC apoptosis upon DSS treatment or exogenous TNF, they do not cross to *Ripk3*^{-/-} or combined *Ripk3*/*Casp8* deficient backgrounds. You would predict, based on the model, that *Ripk3*^{-/-} would not rescue and combined *Ripk3*/*Casp8* deficiency would rescue.
3. Similarly, *Efhd2*^{f/f} *Vil1cre*^{+/+} mice should be rescued by TNFR1 or TNF deletion in vivo, to rule out additional sensitivity to TNF-independent death pathways.
4. Although the authors show that enteroids and colon cancer cells lines are more sensitive to TNF plus CHX or SMAC mimetics, but not to necroptotic stimuli, they do not present data of cells stimulated with TNF alone. Enteroids are generally more sensitive to TNF than colon cancer cell lines,

and there should be some detectable increased sensitivity to TNF alone in *Efhd2*-deficient enteroids. This is important, since in vivo CHX and SMAC mimetics are not present. Also, TNF/CHX or TNF/SMAC mimetic drive toward apoptosis, but it could be that stimulation with TNF alone would demonstrate increased sensitivity to necroptosis as well.

5. Since EFHD2 promotes NF κ B signaling, it could be that EFHD2^{-/-} IECs have lower expression of anti-apoptotic proteins (e.g., cFLIP, IAP proteins, or bcl-2/bcl-xl).

MINOR

1. I recommend adding MW markers to the immunoblots.

Point-to-point responses

To Reviewer #2

The study looks at the mechanism by which Ehd2 suppresses inflammation in intestinal epithelial cells by blocking phosphorylation of Cofilin and inhibiting internalization of TNFR1 to mediate RIPK1-induced apoptosis. The role of TNFR1 in driving RIPK1-mediated apoptosis via activation of caspases in the intestinal epithelium is well-described in the literature. The authors have extended this field by demonstrating a role for Ehd2 in inhibiting phosphorylation of Cofilin in this pathway. While the in vivo experiments are comprehensive, and the overall methodology is well-designed, there are a number of issues that influence the impact and support for the mechanisms proposed, which are outlined below. There is also a need for improvement in the scientific language throughout the paper, citations where indicated, a quantitative analysis of all the immunoblots, and information on the human patient data. Histology and immunostaining images lack representation of controls, and secondary staining controls, that need to be added to the body of the manuscript or to supplementary data. Overall, the study is sound and contributes to the advancement of the field of intestinal epithelial cell death signalling.

Response: Many thanks for the positive comments. With new data (new **Fig. 1a-c, g; Fig. 2d, e, h, i; Fig. 3b-e; Fig. 4b-d; Fig. 7g; Supplementary Fig. 2c; Supplementary Fig. 3i, j; Supplementary Fig. 5a-f; Supplementary Table 1**) and corresponding revisions, we believe we have addressed the concerns in the revised manuscript and further improved the quality of this work.

Major Comments:

Comments on figures:

Figure 1a

- Details on Control and UC patients are missing. The authors should consider including the median age, gender, medical history if available and region of the colon where the tissue was obtained from.

Response: We have provided the details of sample information on UC patients and controls in **Supplementary Table 1**.

- A quantification of Efhd2 should be included for the immunohistochemistry images from control and UC patients to determine if there were any differences in the staining intensity or patterns of Efhd2 localization across the patient cohort.

Response: According to the suggestion, we adjusted the concentration of primary antibody for immunohistochemistry staining of EFHD2 to reduce nonspecific staining and refined the quantitative analysis (**Fig. 1a, c**). The immunostaining showed that EFHD2 was mainly expressed in the epithelium of colon tissues, with relatively lower expression in the lamina propria across the patient cohort. The EFHD2 protein expression in intestinal biopsy samples was significantly decreased in UC patients at the active stage compared with that in UC patients at the remission stage and in healthy controls according to the histologic scores (**Fig. 1c**).

- While the paper is based on a cell death signalling pathway in an in vivo colitis model, the authors do not include any discussion pertaining to IBD models or the use of anti-TNF therapy in UC patients in the discussion section. This needs to be addressed.

Response: Thanks for the comment. Since our study indicated EFHD2's role in inhibiting TNF-induced IEC apoptosis, we therefore examined whether EFHD2 expression was correlated with UC patients' responsiveness to anti-TNF therapy. We further performed EFHD2 immunostaining of intestinal biopsies from responders and non-responders of UC patients receiving anti-TNF therapy (Infliximab treatment). The results showed that the expression level of EFHD2 in the intestinal epithelium of

responders was higher than that of non-responders, indicating higher EFHD2 expression might be correlated with better response to anti-TNF therapy in UC patients (**Fig. 7g**). Our results suggested EFHD2 may be a potential predictor to guide the clinical treatment of ulcerative colitis with anti-TNF therapy. This issue has been addressed in the Discussion section.

- Figure 1a will also benefit from a subset image showing a magnified view of Efh2 localization to epithelial cells (and immune cells, too?). A secondary control for the antibody should be included in the supplementary data, as the staining appears quite ubiquitous here.

Response: Per reviewer's suggestion, we adjusted the concentration of primary antibody for immunohistochemistry staining of EFHD2 to reduce nonspecific staining and added the secondary Ab control as well as the magnified images of EFHD2 staining (**Fig. 1a**). The magnified view of the immunostaining showed that EFHD2 was mainly expressed in the epithelium of colon tissues, with relatively lower expression in the immune cells of the lamina propria across the patient cohort (**Fig. 1a**).

- Did the authors see any decrease in TNFR1 or caspases corresponding with decreased Efh2 in the UC tissues compared to those in remission or in controls?

Response: As reviewer suggested, we detected the expression level of caspase-3 and TNFR1 by immunohistochemistry in both colonic biopsies of UC patients and healthy controls. Our results showed the level of caspase-3 was lower in UC patients at the active stage compared with that at the remission stage and in healthy controls (Below, Figure 1 for Reviewer #2). It has been reported that the decreased level of caspase-3 in biopsies from patients with IBD was associated with the reduced expression of the tight junction protein occludin (Kuo, W.-T. et al. *Inflammation-induced Occludin Downregulation Limits Epithelial Apoptosis by Suppressing Caspase-3 Expression. Gastroenterology* 157, 1323–1337 (2019)). Despite we observed the similar variation trends for EFHD2 expression in

colonic biopsies of UC patients, it is still hard to prove the casual link between caspase-3 and EFHD2 expression. Moreover, the elevated apoptosis as indicated by increased cleaved caspase-3 is also widely reported in the intestinal epithelium of IBD patients (Patankar, J. V. & Becker, C. *Cell death in the gut epithelium and implications for chronic inflammation. Nat. Rev. Gastroenterol. Hepatol.* 17, 543–556 (2020); Nunes, T., Bernardazzi, C. & de Souza, H. S. *Cell Death and Inflammatory Bowel Diseases: Apoptosis, Necrosis, and Autophagy in the Intestinal Epithelium. BioMed Res. Int.* 2014, e218493 (2014)). This aligns with the reduced EFHD2 expression at the active stage of UC.

Meanwhile, the level of TNFR1 remained unchanged between the different groups (Below, **Figure 1 for Reviewer #2**), which was consistent with our findings that EFHD2 restricted the internalization of TNFR1 during colitis, without affecting the level of TNFR1, to inhibit intestinal inflammation.

Figure 1 for Reviewer #2. Representative images of immunostaining for Caspase-3 (n = 3) and TNFR1 (n = 4) on intestinal biopsy samples from UC patients at the active or the remission stage, and healthy controls. Histologic scores of Caspase-3 and TNFR1 staining were using a 3-point quantification scale. Error bars show means ± s.d. *P < 0.05, **P < 0.01, and ***P < 0.001, two-tailed unpaired Student’s t test.

Figure 1b

- Instead of a n=2 representation, this figure would benefit from a subset image in the lower panel showing a magnified view of Efhd2 localization to epithelial cells and immune cells with the secondary controls.

Response: Thanks for the suggestion. We adjusted the concentration of primary antibody for immunohistochemistry staining of Efhd2 to reduce nonspecific staining and added the secondary Ab control as well as the magnified images of Efhd2 staining (**Fig. 1b**). The magnified view of the immunostaining showed that Efhd2 was mainly expressed in the epithelium of mouse colon tissues, with relatively lower expression in the immune cells of the lamina propria during DSS-induced colitis (**Fig. 1b**).

- Does Efhd2 localize in a similar manner or to the same levels of expression in different regions of the mouse and human intestine? This is significant, since the authors have focussed on small intestine, enteroids, or the colon for different parts of the study.

Response: We appreciate your suggestion. Since ulcerative colitis and DSS-induced experimental colitis mainly cause pathological changes in the colon, we firstly examined the level of Efhd2 expression in the small intestine of C57BL/6J mice at steady state or mice administrated with recombinant TNF by immunohistochemistry staining. We found comparable expression as well as the similar localization of Efhd2 in the small intestine or colon of mice before and after treatment of TNF at 4h (**Below, Figure 2 for Reviewer #2**).

However, considering TNF-induced IEC death was as rapid as 4h which was much faster than DSS-induced colitis, we speculated that the expression pattern of Efhd2 in the TNF-induced colitis model was different from that of DSS-induced colitis. Therefore, the decreased level of EFHD2 in patients with UC or in mouse model of DSS-induced colitis might be not dependent on TNF signaling. It will need further

exploration to examine the expression pattern and localization of EFHD2 in patients with Crohn's disease or in other models of enteritis in the future.

Figure 2 for Reviewer #2. Representative images of immunofluorescence staining for Efh2 of the small intestine and colon from C57BL/6J mice before and after i.v. injection with recombinant TNF for 4h.

Figure 1f

- Figure labels indicating the timepoint in the DSS experiment are missing in this panel. The figure will benefit from showing a D0 control, instead of a n=2 representation and the inclusion of a lower magnification of the image that shows the extent of damage in DSS in the whole intestine.
- Please include the same for Figure 2d and e.

Response: We have added representative images showing controls on day 0, including a lower magnification of the images in **Fig. 1g, 2d, 2e**.

Figure 2

- Fig. 2h, line 142-145, the authors mention an “obvious decrease of IECs and impaired intestinal barrier upon DSS treatment” in Efh2f/fVil1cre/+ mice compared to Efh2f/f controls due to loss of E-cadherin. There is no quantitative measurement of barrier function in these mice, so the conclusion

needs to be more circumspect or a functional measurement of the barrier should be included.

Response: Following the suggestion, we performed the intestinal permeability assay, which showed higher permeability of the intestinal barrier in *Efhd2*^{f/f} *Vil1*^{cre/+} mice than that in *Efhd2*^{f/f} mice upon DSS treatment (**Fig. 2i**). The data suggested a severely damaged intestinal barrier, corresponding to the results in **Fig. 2h**.

Figure 3

- A magnified subset image showing the specific localization of Casp3, 8 and p-MLKL should be included in Fig.3b.

Response: The original immunofluorescent images of cleaved caspase-3, cleaved caspase-8, and p-MLKL were replaced with magnified subset images (**Fig. 3b**).

- Considering the ubiquitous expression of *Efhd2* on epithelial cells, is the protection conferred limited only to surface epithelial cells or throughout the crypt axis? Fig.3b shows that deletion of *Efhd2* in the epithelium results in significant cell death only in the surface epithelium. Can the authors include a comment on it or further analysis?

Response: In our study, an elevated rate of IEC apoptosis was observed in the *Efhd2*^{f/f} *Vil1*^{cre/+} mice compared to the control group, which was primarily localized to the top regions of villi in the intestinal epithelium from DSS-treated mice, while a small portion occurred within the midsection of the crypt axis (**Fig. 3b**). Further detailed description on these findings have been included in the Results section.

- The immunoblot showing p-MLKL and cleaved caspase 3 activity should be quantified and analysed statistically to show the differences in the N numbers.

Response: The immunoblot showing p-MLKL, and cleaved caspase-3/8 activity have been quantified to statistically analyze the difference between WT and *Efhd2*^{-/-} IECs (**Fig. 3d**).

- Fig. 3e, does not mention the timepoint of the DSS experiment used for Bulk RNA-Seq analysis. This should be clearly indicated in the Figure legend.

Response: Sorry for the mistake. Fig. 3e showed the bulk RNA-seq analysis of IECs from DSS-treated *Efhd2^{f/f} Vil1^{cre/+}* and *Efhd2^{f/f}* mice on day 7. We have corrected it in the figure legend.

Figure 4

- Were the doses of TNF, Cycloheximide and LCL161 determined from a dose-curve in vitro or in vivo or as a guideline from literature? Rationale needs to be included for the choices.

Response: Sorry for the missing. The corresponding dosage was determined based on guidance from the literature, which has been included in Method section.

- Did the authors see significant blunting of villi and crypts in *Efhd2^{f/f} Vil1^{cre/+}* animals compared to *Efhd2^{f/f}* mice in Fig. 4b? This panel will benefit from the inclusion of a low magnification image and a quantification of the blunted villi and crypts in the small intestine. Did the authors see any changes in the crypt length of the colon of *Efhd2^{f/f} Vil1^{cre/+}* mice?

Response: We have performed experiments in Figure 4 at least three times to ensure compliance with the requirement for three independent biological replicates. Our results showed shortened villi and more severe damage in the intestinal barrier in the small intestine of *Efhd2^{f/f} Vil1^{cre/+}* mice after TNF treatment for 4h (**Fig. 4b**). We have included the low magnification images and quantification of villus height in **Fig. 4b**. However, there was no significant difference in the villus height of the colon between *Efhd2^{f/f} Vil1^{cre/+}* and *Efhd2^{f/f}* mice after TNF treatment (**Below, Figure 3 for Reviewer #2**), because TNF-induced rapid IEC death mainly appeared at the villi tips of the small

intestine (Zhou, L. et al. Group 3 innate lymphoid cells produce the growth factor HB-EGF to protect the intestine from TNF-mediated inflammation. *Nat. Immunol.* 23, 251–261 (2022)).

Figure 3 for Reviewer #2. Representative images and quantification of villus height of H&E staining of the colon from *Efh2^{f/f}* and *Efh2^{f/f} Vill1^{cre/+}* mice after i.v. injection with recombinant TNF for 4h. Error bars show means \pm s.d. * $P < 0.05$, ** $P < 0.01$, and *** $P < 0.001$, two-tailed unpaired Student's t test.

- Is there a reason the authors did not include the TNF only controls for the enteroids experiments in Fig. 4c and d? At what point during organoid growth were TC or TS added to the cultures? The enteroids at T0 appear be immature (little to no budding) and significantly different in terms of size for WT vs *Efh2^{-/-}*.

Response: Thanks for your suggestion. We improved the organoid culture technique, and TNF, TS or TC stimulation was initiated on the fifth day of culture when the enteroids became mature. Representative images of WT and *Efh2^{-/-}* enteroids cultured from day 2 to day 10 was updated in **Supplementary Fig. 5a**. The viability and buds on day 5 were similar between WT and *Efh2^{-/-}* enteroids (**Supplementary Fig. 5b, c**). We subsequently performed TNF-induced apoptosis experiments on mature enteroids. Live imaging and flow cytometry analysis demonstrated increased caspase-3/7 activity in *Efh2^{-/-}* enteroids than in control enteroids treated with TNF for 24h (**Fig. 4c, d**).

Similar findings were also observed in enteroids treated with TS or TC (Supplementary Fig. 5e, f).

- Although the *Efhd2*^{-/-} organoids +TC/TS appear to be highly apoptotic, they also seem much bigger than controls. Have the authors determined if *Efhd2* influences proliferation, or its loss causes an increase in proliferation and budding in organoids or in the mouse native tissue?

Response: Following to the reviewer's comment, we addressed this issue by detecting the development of WT and *Efhd2*^{-/-} enteroids. We performed immunostaining for Edu or markers for Paneth cell (Lysozyme) and goblet cell (Muc-2), and observed the similar development, differentiation and proliferation between WT and *Efhd2*^{-/-} enteroids (Supplementary Fig. 5d). In addition, we conducted the similar staining in intestinal tissues including Ki67 to detect the proliferation of intestinal epithelium, and multiple markers of different IEC subtypes of small intestine and colon, such as Lysozyme for Paneth cells, Chromagranin-A for enteroendocrine cells, CAI for colonocytes, Muc-2 and PAS for goblet cells. Indeed, we found that *Efhd2* deficiency in IECs did not affect intestinal development, differentiation and proliferation (Supplementary Fig. 3i, j).

- Figure 5 and 6 requires the immunoblots to be quantified with statistics.

Response: We performed quantification analysis of all immunoblots in below figures (Figure 4, Figure 5 for Reviewer #2, relative to Fig. 5, 6).

Figure 4 for Reviewer #2. Quantification analysis of all immunoblots in Fig. 5.

Figure 5 for Reviewer #2. Quantification analysis of all immunoblots in **Fig. 6**.

- Supplementary Figures: It would be beneficial to see immunohistochemistry staining for *Efhd2* in the *Vil1Cre/+* and *Efhd2^{-/-}* mice, relative to its controls.

Response: Done. We performed immunohistochemistry staining of *Efhd2* in WT and *Efhd2^{-/-}* mice (**Supplementary Fig. 2c**) as well as in *Efhd2^{f/f}* and *Efhd2^{f/f} Vil1^{cre/+}* mice (**Supplementary Fig. 3g**) to confirm the deletion of *Efhd2* in mice intestinal epithelium.

- Were there any inherent differences in the development and differentiation of epithelial lineages in either the enteroids or the native epithelium of *Efhd2^{f/f}Vil1cre/+* mice compared to *Efhd2^{f/f}* controls?

Response: We performed immunostaining for Edu or markers for Paneth cell (Lysozyme) and goblet cell (Muc-2), and observed the similar development, differentiation and proliferation between WT and *Efhd2^{-/-}* enteroids (**Supplementary Fig. 5d**). Likewise, by Ki67 immunostaining in the epithelium of *Efhd2^{f/f}* and *Efhd2^{f/f} Vil1^{cre/+}* mice to detect the proliferation of intestinal epithelium, and staining for multiple markers of different IEC subtypes of small intestine and colon, such as Lysozyme for Paneth cells, Chromagranin-A for enteroendocrine cells, CAI for colonocytes, Muc-2 and PAS for goblet cells, we found that *Efhd2* deficiency in IECs

did not affect intestinal development, differentiation and proliferation (**Supplementary Fig. 3i, j**).

General minor comments on the paper

- The conclusion statement in the abstract needs to be re-written to align with the data presented.

Response: We have re-written the conclusion statement in the Abstract.

- Line 41, IEC- define abbreviations when used for the first time.

Response: We have corrected this sentence.

- Line 55, TNFR1 cell death signalling should be corrected to TNFR1-mediated cell death signalling.

Response: We have corrected this sentence.

- Line 56, the spelling of “translocation” should be corrected.

Response: We have corrected it.

- The formatting of the mouse genotypes in the manuscript should be consistent.

Response: We have corrected it.

- Line 133, there is no quantification to reflect the differences that justify changes in intestinal development and differentiation in Supplementary Fig. 3i.

Response: According to the suggestion, we have added the quantification analysis to statistically analyze the difference in intestinal development and differentiation between *Efhd2*^{f/f} *Vil1*^{cre/+} and *Efhd2*^{f/f} mice. We found that *Efhd2* deficiency in IECs

did not affect intestinal development, differentiation and proliferation (**Supplementary Fig. 3i, j**).

- Line 156, 228, 303, 335- include a citation.

Response: We have included the citation correspondingly.

- Line 161, include a citation for p-MLKL as a classical marker of necroptosis.

Response: We have included the citation correspondingly.

- Several sentences require revision for clarity, see for example- Line 171,179, 187, 195, 197, 222, 363, 368 and 380-385.

Response: We have corrected it correspondingly.

- Line 211-212, both cells should be corrected to both "cell lines".

Response: We have corrected it.

- Line 222, "EFHD2-deficiency cells" should be corrected to "EFHD2-deficient cells".

Response: We have corrected it.

- Line 307-311 is a complicated sentence – it should be revised for clarity and precision.

Response: We have revised this sentence.

- Line 321, rewrite "we report that..." since it is not a part of this study, it was a previous study done by the group.

Response: We have corrected this sentence.

- Line 329, "cause" should be corrected to "causes".

Response: We have corrected it.

- Line 456, a single cell suspension cannot be obtained through a 70µm filter, it requires a 40µm filter.

Response: We have added the details in Method section.

- If the concentrations of CHX and LCL161 were determined from literature, cite references.

Response: Sorry for the missing. The corresponding dosage was determined based on guidance from the literature, which has been included in Method section.

To Reviewer #3

In their manuscript "EFHD2 Suppresses Intestinal Inflammation by Blocking Intestinal Epithelial Cell TNFR1 Internalization" Wu et al. describe a role for EFHD2 as a negative regulator of IEC death upon TNF-stimulation. EFHD2 deficiency, as it is found in UC, leads to aberrant IEC death and associated intestinal inflammation while its presence can protect IECs from TNF-induced apoptosis.

Major comments

While several dysregulated pathways are responsible for the aberrant and continuous inflammation in IBD, the findings of this paper describe an interesting further mechanism that drives inflammation in UC. Overall, the paper is well written and the experiments performed are appropriate.

Response: Thanks for the positive comments. With new data (new **Fig. 1b, g, h; Fig. 3b, d; Fig. 4c, d; Fig. 5a, e, f; Fig. 6c, f, g; Fig. 7g; Supplementary Fig. 1a; Supplementary Fig. 2b, f, g; Supplementary Fig. 5b-g; Supplementary Fig. 6a-c, g; Supplementary Fig. 7c, d**) and corresponding revisions, we believe we have

addressed the concerns in the revised manuscript and further improved the quality of this work.

The authors use organoid cultures to examine TNF-induced RIPK1-dependent and -independent apoptosis. However, the functionality and viability of the differentiated enteroids need to be confirmed in EFHD2^{-/-} as well as wt enteroids. Please perform functional assays as well as stainings for enteroid markers. Also, please perform stainings for necrotic cores to exclude their formation within the enteroids. In addition to confirming the functionality and viability of enteroids, this would also help to exclude any differences in the development and functionality between EFHD2^{-/-} and wt enteroids.

Response: We appreciate your suggestion. To confirm the functionality and viability of the differentiated enteroids in WT and *Efhd2*^{-/-} enteroids, we conducted a series of experiments. First of all, the viability and buds were similar between WT and *Efhd2*^{-/-} enteroids (**Supplementary Fig. 5b, c**). Moreover, by staining for Edu or markers for Paneth cell (Lysozyme) and goblet cell (Muc-2), *Efhd2*^{-/-} enteroids showed similar development, differentiation and proliferation compared with WT enteroids (**Supplementary Fig. 5d**). In addition, we performed HOECHST and PI staining in WT and *Efhd2*^{-/-} enteroids for necrotic cores according to the literature (*Watanabe, M. et al. Self-Organized Cerebral Organoids with Human-Specific Features Predict Effective Drugs to Combat Zika Virus Infection. Cell Rep. 21, 517–532 (2017)*). Live imaging indicated that no necrotic cores (PI-positive core) was found within the enteroids (**Below, Figure 1 for Reviewer #3**), which was consistent with the results showing in **Supplementary Fig. 5g**.

Figure 1 for Reviewer #3. Representative live cell images of WT and *Efh2*^{-/-} enteroids cultured on day 5 in the presence of PI (red) and HOECHST (blue).

Fig. 5: Why was TNF stimulation performed in HCT116 cells and TNF+CHX and TNF+LCL-161 stimulation as well as necroptosis-induction in HT29 cells? If all experiments are performed in both cell lines, for consistency please show results for one cell line in the main figures and the results for the other cell line in the supplementary section. This also applies to Fig. 6. In Fig. 5a caspase 3 expression is very inconsistent over the time measured and between wt, siRNA1 and siRNA2. (b) Caspase 8 cleavage was measured in the HT29 stimulation experiments but not in the HCT116 experiments; please explain why. Also, please label the kDa of the proteins cleaved and uncleaved in all western blot pictures.

Response: Thanks for the suggestion. However, we found that the induction of apoptosis in HT-29 cell line was insignificant when treated with TNF alone, according to our preliminary experiment and the literature (*Chopra, A. et al. Efficient Activation of Apoptotic Signaling during Mitotic Arrest with AK301. PLOS ONE 11, e0153818 (2016)*). Therefore, we induced apoptosis in HT-29 cell line treated with TNF in combination with CHX or LCL-161 (TNF-induced cell-death checkpoint inhibitor), corresponding to the guideline of literature (*Patankar, J. V. et al. E-type prostanoid receptor 4 drives resolution of intestinal inflammation by blocking epithelial necroptosis. Nat. Cell Biol. 23, 796–807 (2021)*); *Du, J. et al. RIPK1 dephosphorylation and kinase activation by PPP1R3G/PP1 γ promote apoptosis*

and necroptosis. *Nat. Commun.* 12, 7067 (2021)). Meanwhile, necroptosis couldn't be induced in the HCT-116 cell line, which is lack of RIPK3, by TNF alone and TNF combination with either CHX or LCL-161 (Negroni, A. et al. *RIP3 AND pMLKL promote necroptosis-induced inflammation and alter membrane permeability in intestinal epithelial cells. Dig. Liver Dis.* 49, 1201–1210 (2017)). That's the reason why we didn't perform all the experiments in both cell lines.

For consistency, we performed Western blot analysis of lysates from HCT-116 cells after knockdown of *EFHD2* then treated with TNF alone, TC or TS for the indicated time (**Fig. 5a**), corresponding to TC or TS stimulation in HT-29 cell line (**Supplementary Fig. 6c**). Besides, we performed western blot analysis in WT and *EFHD2*^{-/-} HCT-116 cell line with TNF or TS treatment (**Supplementary Fig. 6g**), corresponding to the TS stimulation in HT-29 cell line (**Fig. 5e**). Our results demonstrated that caspase-3 expression was consistent regardless of *EFHD2* interference in HCT-116 cell line with TNF treatment (**Fig. 5a**).

We have labeled the kDa of the proteins in all Western blot images.

Fig. 5g: The result section states that under TNF treatment phosphorylation of IKKa/b and p65 in *EFHD2*^{-/-} cells was significantly decreased compared to wt. However, Fig. 5g doesn't show any difference in IKKa/b phosphorylation between *EFHD2*^{-/-} cells and wt. Also, IKKa/b expression seems inconsistent over the time it was measured. For p65, phosphorylated p65 is present in *EFHD2*^{-/-} cells as well as wt cells but then phosphorylated p65 decreases faster in *EFHD2*^{-/-} cells compared to wt. Is there a reason why phosphorylated p65 is initially also present in the *EFHD2*^{-/-} cells and/or why it decreases faster in the *EFHD2*^{-/-} cells? Quantification of the proteins of interest could help to support the statement.

Response: To address this concern, we optimized the culture conditions to ensure that HCT-116 cells maintain a steady state before TNF stimulation. We then re-evaluated the NF-κB pathway in WT and *EFHD2*^{-/-} HCT-116 cells. Under TNF stimulation, we

found the significantly decreased phosphorylation of IKK α/β and P65 in *EFHD2*^{-/-} HCT-116 cells compared to controls, with the consistent level of IKK α/β and P65 over the time in both groups (**Fig. 5f**). Besides, phosphorylated P65 was almost absent in both groups at 0h. We performed quantification analysis of the immunoblot in **below figure (Figure 2 for Reviewer #3, relative to Fig. 5f)**.

Figure 2 for Reviewer #3. Quantification analysis of the immunoblot in **Fig. 5f**.

Line 258/Fig. 6f: The main text mentions that there were significantly higher levels of TNFR1 in early endosomes in *EFHD2*^{-/-} HT29 cells compared to wt cells. However, the graph in Fig. 6f doesn't show any significance. Please also perform an n of 3.

Response: We have performed three independent experiments and the quantification (n = 3) was included in **Fig. 6f**.

Minor comments

Considering that *EFHD2* has previously been shown to be upregulated in various cancer tissues and has been associated with metastasis, the therapeutic potential of *EFHD2* must be taken with caution. This is especially the case as the authors found *EFHD2* to be only downregulated during active UC while its expression recovered during remission.

Response: Thanks for the insightful comments. Indeed, our discovery underscores the role of *EFHD2* in inhibiting IEC apoptosis and intestinal inflammation, while upregulated *EFHD2* in cancer cells has been reported to be associated with metastasis

(Moreno-Layseca, P. et al. Cargo-specific recruitment in clathrin- and dynamin-independent endocytosis. *Nat. Cell Biol.* 23, 1073–1084 (2021)). Given the dual role of EFHD2, its therapeutic potential may be constrained. However, we uncovered the potential association between EFHD2 expression level and the responsiveness of the UC patients undergoing anti-TNF therapy (Fig. 7g). As an inhibitor of TNF receptor internalization, EFHD2 expression may concurrently serve as the predictive marker to guide the clinical treatment of ulcerative colitis with anti-TNF therapy.

Line 172: Please change to "It revealed that genes with an increased expression in EFHD2^{-/-} IECs compared to controls were associated with the acute inflammatory response..."

Response: We have corrected this sentence.

Line 545: From which tissue were the endosomes isolated and how much tissue was used for endosome isolation?

Response: We have added details of endosome isolation in the sentence. Endosomes of HT-29 cells (approximately $3\text{-}5 \times 10^7$ cells) were isolated in this experiment.

qPCR: How much RNA was used for RT-PCR/qPCR?

Response: We have added the details in Method section. 1 μg of RNA was reverse transcribed, and 40 ng of cDNA was further used for the qPCR assay.

Figures:

Please label the kDa of the proteins shown on the Western Blots, especially when several bands are visible. What are the other bands that are visible?

Response: We have added all the kDa of the proteins to all the immunoblots.

Fig. 1b: The legend mentions that mice were treated for the indicated times, however, mice were treated for 7d and then left without treatment until d12.

Response: Thanks for the suggestion. We have modified the figure legend in **Fig. 1b** for clarification. C57BL/6J mice treated with 3% DSS for 6 days and left without treatment until day 7 or day 12 in **Fig. 1b**.

Fig. 1f+g: Please mention after how many days of DSS treatment the staining was performed.

Response: We have modified the figure legend in **Fig. 1g, h** for clarification. WT and *Efhd2*^{-/-} mice were administrated with 3% DSS administration for 6 days and sacrificed on day 7. The H&E staining was performed on the colon sections of mice on day 7.

Fig. 2g: This should read "Relative gene expression of *Efhd2*^{f/f} and *Efhd2*^{f/f} 737 *Vil1cre*^{+/+} mice after 7d of DSS treatment compared to d0".

Response: We have corrected this sentence.

Fig. 3b: For completion, please add a zoomed-in picture of p-MLKL as it was done for the other conditions.

Response: We have added the magnified images of p-MLKL for completion (**Fig. 3b**).

Fig. 3d: In Fig. 3b + c Caspase 3, caspase 8 and p-MLKL have been analysed, therefore, the western blot in Fig. 3d should also show Caspase 8.

Response: The immunoblot showing p-MLKL, and cleaved caspase-3/8 activity have been performed and quantified to statistically analyze the difference between WT and *Efhd2*^{-/-} IECs (**Fig. 3d**).

Fig 4: Fig. 4 states that data are representative of at least 2 independent experiments. However, a minimum of 3 independent experiments should be performed.

Response: We have performed experiments in Figure 4 at least three times to ensure compliance with the requirement for three independent biological replicates.

Fig 4c+d show that both treatments, TNF+CHX and TNF+LCL-161, induce an increase of Caspase3/7 in EFDH2^{-/-} enteroids compared to wt. However, in Fig. 4e only TNF+LCL-161 was further analysed; why was TNF+CHX not analysed any further?

Response: We appreciate your suggestion. We improved the organoid culture technique, and TNF, TS or TC stimulation was initiated on the fifth day of culture when the enteroids became mature. Live imaging and flow cytometry analysis revealed the increased caspase-3/7 activity in *Efhd2*^{-/-} enteroids compared to control enteroids treated with TNF for 24h (**Fig. 4c, d**). Similar findings were also observed in enteroids after treatment with TS or TC (**Supplementary Fig. 5e, f**).

Fig. 6 c+d: Please perform a minimum of n=3. Please explain in one sentence why TS was also measured after 0.5 h; is TNFR1 quicker internalised after TS treatment compared to TC treatment? If so, Fig. 6d should be a Suppl. Fig but also include TC treatment after 0.5h.

Response: We performed the analysis after TS or TC treatment on 4h with n = 3 (**Fig. 6c**), and included TC treatment on 30min in **Supplementary Fig. 7c**.

As for the detecting timepoint, we firstly examined the cell-surface level of TNFR1 at 4h after TS or TC stimulation and discovered the significantly decreased surface TNFR1 level in *EFHD2*^{-/-} HT-29 by flow cytometric analysis (**Fig. 6c**). The internalization of TNFR1 began at the early stage after TNF stimulation (*Schütze, S., Tchikov, V. & Schneider-Brachert, W. Regulation of TNFR1 and CD95 signalling by receptor compartmentalization. Nat. Rev. Mol. Cell Biol. 9, 655–662 (2008)*). Therefore, we further detected the cell-surface level of TNFR1 at 30min after TS or TC stimulation. Similarly, we found significant decrease at the surface level of TNFR1 even at the early stage of the receptor endocytosis (30 min) upon TS or TC treatment (**Supplementary Fig. 7c**).

Compared to TC treatment, the internalization of TNFR1 was relatively quicker upon TS treatment (**Supplementary Fig. 7c**).

Fig. 6g: Why was this experiment performed with TNF+LCL161? Would it be sufficient if the experiment was performed with TNF alone?

Response: We pretreated WT and *Efhd2*^{-/-} enteroids with MDC to inhibit TNFR1 internalization, and then treated them with TNF alone. When the internalization of TNFR1 was restrained, we found that the increased apoptosis in *Efhd2*^{-/-} enteroids was counteracted, which was shown by comparable caspase-3/7 activities between WT and *Efhd2*^{-/-} enteroids (**Fig. 6g**), corresponding to the results treated with TS (**Supplementary Fig. 7d**).

Fig. 7g: (Suggestion) An option to present the right-hand side of the graph where the situation in EFHD2 negative cells is described, would be to cross out EFHD2. Instead of drawing an arrow from EFHD2 to cofilin, an arrow around cofilin itself could be drawn to show its self-phosphorylation.

Response: We have reorganized in **Fig. 7g** according to your suggestion.

Suppl. Figures

Suppl. Fig. 1: Comparison of protein expression is difficult if the housekeeper gene is expressed differently. Has the same amount of protein been loaded onto each lane?

Response: The protein loading amount for each lane was determined by BCA assay. The immunoblot in mice tissues have been repeated and quantified to statistically analyze the expression of *Efhd2* across the different organs (**Supplementary Fig. 1a**). Our results showed the relatively higher level of *Efhd2* in intestine and brain (**Supplementary Fig. 1a**).

Suppl. Fig. 2: Please perform a minimum n of 3.

Response: We have performed experiments with $n \geq 3$ in **Supplementary Fig. 2**.

Suppl. Fig 2b: From which tissue was the RNA isolated that was used to measure *efhd2* expression in the created K/O mice?

Response: The RNA was isolated from colon tissues to measure *Efhd2* mRNA level in WT and *Efhd2*^{-/-} mice (**Supplementary Fig. 2b**).

Suppl. Fig 2e+f: To strengthen the data, please perform a minimum of $n=3$.

Response: We have performed CyTOF analysis of the proportion of isolated intestinal immune cells from the lamina propria of WT and *Efhd2*^{-/-} mice before ($n = 3$) and after ($n = 4$) DSS administration on day 7 (**Supplementary Fig. 2f, g**).

Suppl Fig. 5b+c: The main text states that EFHD2 was knocked down in HCT 116 and HT29 cells, however, the figures only show HCT116 cells.

Response: Sorry for the mistake. We have detected the mRNA and protein level of EFHD2 after knockdown in HT-29 cell line and added the results in Supplementary data (**Supplementary Fig. 6a, b**).

Suppl. Fig 5c: What is 0, T and TZ? Please explain in the legend. Why does the amount of EFHD2 protein in NC increase from 0 over T to TZ? Was the relative gene expression in Fig. 5b measured at 0, T or TZ?

Response: Sorry for the mistake. **Supplementary Fig. 6b** showed the Western blot analysis for the protein level of EFHD2 in wild-type and *EFHD2* interfered HCT-116 and HT-29 cells treated with TNF alone or TNF combined with zVAD-fmk for 2h. The level of EFHD2 in both cell lines was consistent regardless of TNF treatment (**Supplementary Fig. 6b**). Relatively, the mRNA level of *EFHD2* in **Supplementary Fig. 6a** was measured at steady state in both cell lines. We have already reorganized the relative figure legend for clarification.

To Reviewer #4

In their paper entitled "EFHD2 Suppresses Intestinal Inflammation by Blocking Intestinal Epithelial Cell TNFR1 Internalization", the authors use a variety of approaches to study how EFHD2 restrains intestinal epithelial cell (IEC) apoptosis by inhibiting TNFR1 internalization.

They present IHC of EFHD2 to demonstrate decreased expression in active UC and in DSS colitis. *Efhd2*^{-/-} mice exhibited increased sensitivity to the DSS colitis model. Conditional deletion of *Efhd2* in IECs, but not myeloid cells, increased sensitivity to DSS colitis. *Efhd2* IEC-deficient mice treated with DSS exhibited increased cleaved caspase-3 (CC3) and CC8, but not p-MLKL, consistent with increased sensitivity to apoptosis but not necroptosis *in vivo*. Exogenous TNF also increased CC3 staining in *Efhd2*^{f/f} *Vil1cre*^{+/+} mice. *Efhd2*-deficient enteroids exhibited increased caspase-3/7 activity upon treatment with TNF and CHX or SMAC mimetic, but not under necroptotic conditions using zVAD-fmk. Knock-down or knock-out of EFHD2 in human colorectal cancer cell lines exhibited similar responses to apoptotic stimuli, and immunoblots showed increased CC3 and CC7 but not pRIPK1 or pMLKL in response to TNF with CHX or SMAC mimetic. They then show decreased pIKK α /B and P65 in EFHD2^{-/-} cell lines treated with TNF, suggesting reduced NF κ B signaling. EFHD2 co-immunoprecipitated with RIPK1, TNFR1, and TRADD, and EFHD2^{-/-} cell lines exhibited reduced surface and increased endosomal TNFR1. Finally, the authors show that EFHD2 binds to and inhibits phosphorylation of Cofilin, blocking TNFR1 internalization, and this can be reversed with an LIMK inhibitor.

There are multiple strengths to this study, including use of conditional villin-Cre mice to study *Efhd2* in IECs, and primary intestinal enteroids. They also present mechanistic studies on how EFHD2 restricts IEC apoptosis via Cofilin and TNFR1 internalization. However, there are multiple major and minor concerns that

would limit the impact of the manuscript. The two most significant are a weak link of relevance to IBD and lack of in vivo rescue.

Response: We appreciate the comments. With new data (new **Fig. 1a-c**, **Fig. 4c-j**; **Supplementary Fig. 5e-g**; **Supplementary Fig. 6i**; **Fig. 7g**) and corresponding revisions, we believe we have addressed the concerns in the revised manuscript and further improved the quality of this work.

MAJOR

1. The data presented are insufficient to support the claim that EFHD2 is significantly lower in UC patients. There are a few representative IHC images presented, but there is no quantitation of the 10 patients per group. Importantly, at the mRNA level, EFHD2 exhibits increased overall expression in mucosal biopsies from patients with UC (<https://premedibd.com/genes.html>). If EFHD2 protein expression is discordant with the gene expression data, those data should be presented. It's not clear this would be a good target in IBD, since it restrains apoptosis through scaffolding effects, so it would be difficult to improve its function. It is also not associated with IBD through GWAS. The link between EFHD2 and UC therefore seems tenuous.

Response: Following the suggestion, we adjusted the concentration of primary antibody for immunohistochemistry staining of EFHD2 to reduce nonspecific staining (**Fig. 1a**). Based on the IHC staining, we refined the relative quantitative analysis of the cohort as shown in (**Fig. 1c**). Consistent with our previous findings, EFHD2 expression in intestinal biopsy samples was significantly decreased in UC patients at the active stage compared with that in UC patients at the remission stage and in healthy controls (**Fig. 1a, c**).

At the mRNA level, we conducted further assessments of *Efhd2* level in DSS-treated mice on day 0, day 7 and day 12, corresponding to **Fig. 1b, c**. Our results demonstrated no significant changes over time during the experimental colitis, as

depicted in below Figure (Figure 1 for Reviewer #4). However, IHC staining of *Efhd2* in Fig. 1b exhibited a similar expression pattern to that observed in human biopsies. We speculate that the inconsistent observations between mRNA and protein levels in mouse colon tissues might be due to post-translational modifications or protein degradation processes in *Efhd2*, according to the similar phenomena reported in the literatures (Chen, G. et al. *Discordant Protein and mRNA Expression in Lung Adenocarcinomas*. *Mol. Cell. Proteomics* 1, 304–313 (2002); Ko, J.-L. et al. *MDM2 mRNA expression is a favorable prognostic factor in non-small-cell lung cancer*. *Int. J. Cancer* 89, 265–270 (2000)). Therefore, we hypothesize that the inconsistency in mRNA and protein expression levels may also be present in human samples, and we will further explore this in the future.

Figure 1 for Reviewer #4. Relative gene expression levels of *Efhd2* from colonic tissues from C57BL/6J mice on day 7 and day 12 after 6-day DSS treatment compared to day 0. n = 6 each group; error bars show means \pm s.d. *P < 0.05, **P < 0.01, and ***P < 0.001 compared with WT mice, two-tailed unpaired Student's t test.

Our findings indicated that EFHD2 restrains apoptosis through scaffolding effects, which might constrain its therapeutic potential. However, we uncovered the association between EFHD2 expression level and the responsiveness of the UC patients undergoing anti-TNF therapy (Fig. 7g). The histologic analysis on intestinal biopsy samples of UC patients indicated that higher expression level of EFHD2 might be correlated with better response to anti-TNF therapy (Fig. 7g). Therefore, as an inhibitor of TNF receptor internalization, EFHD2 in the intestinal epithelium might be regarded as the predictive marker of responsiveness to anti-TNF therapy of ulcerative colitis.

2. While the authors present evidence that *Efhd2*^{f/f} *Vil1*^{cre/+} mice exhibit increased sensitivity to IEC apoptosis upon DSS treatment or exogenous TNF, they do not cross to *Ripk3*^{-/-} or combined *Ripk3*/*Casp8* deficient backgrounds. You would predict, based on the model, that *Ripk3*^{-/-} would not rescue and combined *Ripk3*/*Casp8* deficiency would rescue.

Response: We really appreciate your comment and suggestion. Based on TNF-induced apoptosis experiments in enteroids, we found increased caspase-3/7 activity in *Efhd2*^{-/-} enteroids compared to WT controls treated with TNF alone (**Fig. 4c**). Therefore, we further treated enteroids with TNF combined with GSK'872 (inhibitor for RIPK-3) or Z-IETD-FMK (specific inhibitor for caspase-8). We found the increased cell death of *Efhd2*^{-/-} enteroids indicated by PI which could only be rescued by GSK'872 combined Z-IETD-FMK but not by GSK'872 only (**Fig.4e**), indicating that *Efhd2* would protect enteroids from TNF-induced apoptosis but not necroptosis. We would further cross *Efhd2*^{f/f} *Vil1*^{cre/+} mice with *Ripk3*^{-/-} or *Ripk3*^{-/-}*Casp8*^{-/-} mice to perform rescue experiments with DSS or exogenous TNF treatment *in vivo* to address this issue in the future.

3. Similarly, *Efhd2*^{f/f} *Vil1*^{cre/+} mice should be rescued by TNFR1 or TNF deletion *in vivo*, to rule out additional sensitivity to TNF-independent death pathways.

Response: To rule out additional sensitivity to TNF-independent death pathways *in vivo*, we treated *Efhd2*^{f/f} and *Efhd2*^{f/f} *Vil1*^{cre/+} mice with anti-TNF antibody or control IgG intraperitoneally every 2 days since DSS administration. Compared with IgG group, the symptoms of acute colitis in *Efhd2*^{f/f} *Vil1*^{cre/+} mice were obviously alleviated after treated with anti-TNF antibody, manifested as the fewer body weight loss, lower DAI score, longer colon lengths, and milder histologic damage of intestinal epithelium (**Fig. 4f-j**). Our results confirmed intestinal epithelial *Efhd2* could protect mice from

intestinal inflammation by inhibiting TNF-dependent apoptosis, consistent with our previous findings.

4. Although the authors show that enteroids and colon cancer cells lines are more sensitive to TNF plus CHX or SMAC mimetics, but not to necroptotic stimuli, they do not present data of cells stimulated with TNF alone. Enteroids are generally more sensitive to TNF than colon cancer cell lines, and there should be some detectable increased sensitivity to TNF alone in *Efhd2*-deficient enteroids. This is important, since in vivo CHX and SMAC mimetics are not present. Also, TNF/CHX or TNF/SMAC mimetic drive toward apoptosis, but it could be that stimulation with TNF alone would demonstrate increased sensitivity to necroptosis as well.

Response: As suggested, we performed TNF-induced apoptosis on WT and *Efhd2*^{-/-} enteroids with the treatment of TNF alone, TS or TC. Live imaging and flow cytometry analyses revealed increased caspase-3/7 activity in *Efhd2*^{-/-} enteroids compared to control enteroids following 24 hours of TNF treatment (**Fig. 4c, d**). Similar findings were also observed in enteroids after treatment with TS or TC for 3h (**Supplementary Fig. 5e, f**). Our findings demonstrated that *Efhd2* deficiency promoted TNF-induced apoptosis in the cultured enteroids. We further treated enteroids with TNF combined with GSK'872 or Z-IETD-FMK and found the increased cell death of *Efhd2*^{-/-} enteroids, as indicated by PI staining, could only be rescued by GSK'872 combined Z-IETD-FMK, but not with GSK'872 alone (**Fig.4e**). These results suggested that *Efhd2* plays a protective role in shielding enteroids from TNF-induced apoptosis but not necroptosis, in alignment with our previous findings as presented in **Supplementary Fig. 5g**.

5. Since EFHD2 promotes NFκB signaling, it could be that EFHD2^{-/-} IECs have lower expression of anti-apoptotic proteins (e.g., cFLIP, IAP proteins, or bcl-2/bcl-xl).

Response: According to the suggestion, we detected the levels of NF- κ B targeting genes including cIAPs, cFLIP, Bcl-2 or Bcl-x1, which were also decreased in *EFHD2*^{-/-} HCT-116 cells upon TNF treatment compared to WT control (**Supplementary Fig. 6i**). The results demonstrated that EFHD2 suppressed TNF-induced apoptosis but promoted NF- κ B signaling.

MINOR

1. I recommend adding MW markers to the immunoblots.

Response: We have added MW markers to all the immunoblots.

REVIEWERS' COMMENTS

Reviewer #2 (Remarks to the Author):

The authors have made revisions to most of our initial comments to provide insight into TNFR signaling and regulation in intestinal epithelial cells. A few minor points remain: 1) In Fig. 1a, the expression of *Efhd2* appears ubiquitous, it isn't clear from the legend what the arrows are indicating in the panel. 2) there is also a clear difference in the localization of *Efhd2* in d7 of colitis - appears diffuse (or cytoplasmic) rather than membrane as in the other time points. Is this correct or a sampling problem? If correct, should comment, if sampling then replace with more representative image, and 3) the quantification of immunoblots needs to be available to the readers with significance noted. This information isn't shown except in the immunoblot quantification as numbers on top of the blot in the revision statement, but they should be included in the figure as a graph with statistic information available to readers.

The last sentence of the abstract is not very informative of the reported work.

Reviewer #3 (Remarks to the Author):

The authors addressed all concerns, giving more strength to their study.

Regarding the organoid culture, the authors provide an image of wt vs. *EFHD2*^{-/-} organoids stained for PI and Hoechst on d5 to exclude necrosis within the organoids. To support this image, please provide a positive control for PI to confirm that the staining is working considering not a single PI positive cell can be seen in the image.

The authors gave a satisfying explanation, supported by the literature, as to why some experiments in Fig 5 were performed in HCT 116 and others in HT 29 cells.

The authors repeated the experiment in Figure 5f after optimising culture conditions to ensure HCT116 maintained a steady state before TNF stimulation which resulted in more consistent results that now support the statement. Please explain shortly how the culture conditions were improved.

Reviewer #4 (Remarks to the Author):

In their revision, the authors have quantified *EFHD2* staining of patient samples. They also provide demographic information for their cohort. They now demonstrate discordance between RNA and protein levels of *Efhd2* in murine samples, and linked *EFHD2* protein levels to anti-TNF response in a small number of patients. The revised manuscript now includes rescue of *Efhd2*^{f/f}*Vil1cre*⁺ mice with anti-TNF in vivo, and rescue of TNF-induced death in enteroids in vitro using combined pharmacologic inhibition of RIPK3 and Casp8.

The authors have addressed my concerns. One minor point, the figure legend for Figure 7 says "responsible" and "non-responsible" instead of responsive or non-responsive.

Point-to-point responses

To Reviewer #2

The authors have made revisions to most of our initial comments to provide insight into TNFR signaling and regulation in intestinal epithelial cells.

Response: Thanks for your comments. With new or updated data (**Fig. 1a, b; Fig. 5a, f, e; Fig. 6a, b; Fig. 7b, c**) and corresponding revisions, we believe we have addressed the concerns in the revised manuscript and further improved the quality of this work.

A few minor points remain:

1) In Fig. 1a, the expression of Efhd2 appears ubiquitous, it isn't clear from the legend what the arrows are indicating in the panel.

Response: According to the suggestion, we have redefined the regions in the original images, designating the representative region of intestinal epithelium with a dashed box indicated by solid arrows and the representative region of intestinal lamina propria marked by hollow arrows. These annotations illustrated the localization of EFHD2 in different intestinal tissues respectively, which indicated that EFHD2 is mainly expressed in the intestinal epithelium, with relatively lower expression in the lamina propria across the patient cohort (**Fig. 1a; Figure 1 for Reviewer #2, below**).

Figure 1 for Reviewer #2. Representative images of immunostaining for EFHD2 on intestinal biopsy samples from UC patients at the active or the remission stage, healthy controls and controls stained only with the secondary antibody. Scale bar, 200 μ m. Solid arrowheads show intestinal epithelium and hollow arrowheads show the lamina propria.

2) there is also a clear difference in the localization of Efh2 in d7 of colitis - appears diffuse (or cytoplasmic) rather than membrane as in the other time points. Is this correct or a sampling problem? If correct, should comment, if sampling then replace with more representative image.

Response: We appreciate your suggestion. We scanned high-resolution images of all the samples. From the magnified view, two pathologists suggested that EFHD2 was mainly expressed in the cytoplasm, as well as the brush border during DSS-induced colitis. The brush border, characterized by actin-supported membrane protrusions known as microvilli, covers the surface of colonic epithelial cells (*Crawley, SW. et al. Shaping the intestinal brush border. J. Cell Biol. 207, 441-51 (2014)*). Therefore, we suggest that this is not a sampling problem (**Fig. 1b; Figure 2 for Reviewer #2, below**).

Figure 2 for Reviewer #2. Representative images of immunostaining for Efh2 on distal colon tissues from C57BL/6J mice treated with 3% DSS for 6 days and sacrificed at the indicated time with secondary antibody controls. Scale bar, 200 μ m. Solid arrowheads show the brush border.

3) the quantification of immunoblots needs to be available to the readers with significance noted. This information isn't shown except in the immunoblot quantification as numbers on top of the blot in the revision statement, but they should be included in the figure as a graph with statistic information available to readers.

Response: As suggested by the reviewer, we performed quantification analysis of the immunoblot with significance noted in the figures (**Fig. 5a, f, e; Fig. 6a, b; Fig. 7b, c**).

The last sentence of the abstract is not very informative of the reported work.

Response: We have re-written the last sentence in the Abstract.

To Reviewer #3:

The authors addressed all concerns, giving more strength to their study.

Response: Thanks for your comments.

Regarding the organoid culture, the authors provide an image of wt vs. *Efhd2*^{-/-} organoids stained for PI and Hoechst on d5 to exclude necrosis within the organoids. To support this image, please provide a positive control for PI to confirm that the staining is working considering not a single PI positive cell can be seen in the image.

Response: As suggested by the reviewer, we conducted HOECHST and PI staining in WT and *Efhd2*^{-/-} enteroids to assess the presence of necrotic cores. We treated enteroids with TNF combining zVAD-fmk and CHX (TCZ) for 16h as the positive control. Live imaging revealed the absence of necrotic cores (PI-positive core) within the enteroids, compared with the positive control (**Below, Figure 1 for Reviewer #3**).

Figure 1 for Reviewer #3. Representative live cell images of WT and *Efhd2*^{-/-} enteroids cultured on day 5 in the presence of PI (red) and HOECHST (blue). Enteroids treated with TCZ for 16h as the positive control. Scale bar, 50 μ m.

The authors gave a satisfying explanation, supported by the literature, as to why some experiments in Fig 5 were performed in HCT 116 and others in HT 29 cells.

The authors repeated the experiment in Figure 5f after optimising culture conditions to ensure HCT116 maintained a steady state before TNF stimulation which resulted in more consistent results that now support the statement. Please explain shortly how the culture conditions were improved.

Response: We reduced the digestion time of HCT-116 cells to minimize cell damage during the digestion. Experiments were performed once the cells had fully adhered to the walls and exhibited morphological stretching after the HCT-116 cell passaged for 24 hours.

To Reviewer #4:

In their revision, the authors have quantified EFHD2 staining of patient samples. They also provide demographic information for their cohort. They now demonstrate discordance between RNA and protein levels of Efhd2 in murine samples, and linked EFHD2 protein levels to anti-TNF response in a small number of patients. The revised manuscript now includes rescue of *Efhd2^{f/f}/Vil1^{cre}/+* mice with anti-TNF in vivo, and rescue of TNF-induced death in enteroids in vitro using combined pharmacologic inhibition of RIPK3 and Casp8.

The authors have addressed my concerns. One minor point, the figure legend for Figure 7 says "responsible" and "non-responsible" instead of responsive or non-responsive.

Response: We appreciate your positive comments and careful reading. We have corrected the corresponding word in the figure legend.